# Corneal Sensory Receptors and Pharmacological Therapies to Modulate Ocular Pain

**DOI:** 10.3390/ijms26104663

**Published:** 2025-05-13

**Authors:** Ryan Park, Samantha Spritz, Anne Y. Zeng, Rohith Erukulla, Deneb Zavala, Tasha Merchant, Andres Gascon, Rebecca Jung, Bianca Bigit, Dimitri T. Azar, Jin-Hong Chang, Elmira Jalilian, Ali R. Djalilian, Victor H. Guaiquil, Mark I. Rosenblatt

**Affiliations:** 1Department of Ophthalmology and Visual Sciences, Illinois Eye and Ear Infirmary, College of Medicine, University of Illinois Chicago, Chicago, IL 60612, USA; rpark31@uic.edu (R.P.); sspri@uic.edu (S.S.); dzavala3@uic.edu (D.Z.); bbigit2@uic.edu (B.B.); jalilian@uic.edu (E.J.);; 2Jesse Brown Veterans Affairs Medical Center, Chicago, IL 60612, USA; 3Richard and Loan Hill Department of Bioengineering, University of Illinois Chicago, Chicago, IL 60607, USA

**Keywords:** nociceptor, nerve, corneal pain, TRPV1, TRPM8

## Abstract

Nociceptors respond to noxious stimuli and transmit pain signals to the central nervous system. In the cornea, the nociceptors located in the most external layer provide a myriad of sensation modalities. Damage to these corneal nerve fibers can induce neuropathic pain. In response, corneal nerves become sensitized to previously non-noxious stimuli. Assessing corneal pain origin is a complex ophthalmic challenge due to variations in its causes and manifestations. Current FDA-approved therapies for corneal nociceptive pain, such as acetaminophen and NSAIDs, provide only broad-acting relief with unwanted side effects, highlighting the need for precision medicine for corneal nociceptive pain. A few targeted treatments, including perfluorohexyloctane (F6H8) eye drops and Optive Plus (TRPV1 antagonist), are FDA-approved, while others are in preclinical development. Treatments that target signaling pathways related to neurotrophic factors, such as nerve growth factors and ion channels, such as the transient receptor potential (TRP) family or tropomyosin receptor kinase A, may provide a potential combinatory therapeutic approach. This review describes the roles of nociceptors in corneal pain. In addition, it evaluates molecules within nociceptor signaling pathways for their potential to serve as targets for efficient therapeutic strategies for corneal nociceptive pain aimed at modulating neurotrophic factors and nociceptive channel sensitivity.

## 1. Introduction

Nociceptors (from Latin nocere “to harm or hurt”; lit. “pain receptor”) are sensory neurons that respond to damaging or potentially damaging stimuli and are responsible for pain perception [1]. In the skin, nociceptors are composed of free nerve endings of unmyelinated C fibers and lightly myelinated A-delta fibers. A-delta fibers mediate initial fast-onset pain, while C fibers mediate slowly developing “second pain”. Both A-delta and C fibers respond to thermal, mechanical, and chemical stimuli [1]. Mechanonociceptors respond to mechanical stimuli, such as pressure, stretching, and tissue distortion; thermonociceptors detect temperature changes, including thermal stimuli associated with inflammation or tissue damage; and chemonociceptors respond to chemical irritants released by damaged tissues, such as inflammatory mediators, cytokines, and neuropeptides.

When nociceptive neuronal fibers are activated by a stimulus with adequate duration and amplitude, an action potential is produced in the peripheral fiber, propagating through the nociceptor axon to the central nervous system (CNS). The brain interprets the signal, leading to pain perception and physiological and behavioral responses to prevent further harm. Nociceptors are essential to the body’s defense system, detecting environmental changes associated with threats. This pain perception triggers protective responses like reflexive withdrawal, guarding behaviors, and inflammation, all aimed at preventing further damage. In the eye, alteration of nociceptor function can lead to chronic ocular pain conditions and neuropathic pain syndromes secondary to trigeminal nerve damage. In these diseased states, pain can persist even after the initial injury has healed.

To better address corneal pathologies involving nociception, the underlying mechanisms must be understood, which will allow for the development of new corneal antinociceptive therapeutics. Herein, we summarize the key ion channels and neurotransmitters involved in corneal nociception and describe current and potential therapeutics for addressing corneal pain.

## 2. Cornea Nerve Anatomy, Physiology, and Development

Transient receptor potential (TRP) channels act within the context of corneal nociceptive neurons; therefore, it is essential to understand the structure, function, and development of sensory afferent corneal nerves to analyze corneal pain comprehensively. The cornea is innervated by ophthalmic branches of the trigeminal nerve (CN V1). Corneal nociceptors transmit pain via the ophthalmic branch of the trigeminal nerve to neurons in the caudal trigeminal nucleus (Figure 1, Guerrero-Moreno, 2020 [2]). The central branch of the trigeminal nerve travels to the trigeminal brainstem sensory complex in the pons, while the peripheral branch that innervates the ipsilateral cornea travels into the orbit through the ophthalmic branch of the trigeminal ganglion to provide sensory innervation to the ipsilateral cornea [3]. The sensory corneal nerve fibers originate from stromal nerves that form the sub-basal plexus, which groups into radial nerve bundles in the stroma and then penetrate Bowman’s layer before branching into a superficial network in the corneal epithelium (Figure 2 [4]). Innervation of the cornea is organized into four levels, from the penetrating stromal nerve trunks to the intraepithelial nerve terminals. The stromal layer mainly functions as a transit route. The subepithelial layer is particularly dense in the peripheral cornea and feeds axons to the next layer, the sub-basal layer. The sub-basal nerve plexus gives rise to intraepithelial terminals for guiding epithelial migration. Finally, the intraepithelial nerve terminals, which are found throughout the epithelial layers, are the major players in nociception [5]. Corneal nerve fibers can respond to many stimuli (temperature, touch, etc.) [6]. Transduction of these stimuli is mainly achieved via TRP channels located in the corneal nerve fibers. For nociception in the cornea, TRPV1 plays an essential role in pain generation; it has been shown in vitro that various irritants that cause eye pain induce an influx of Ca^2+^ through TRPV1 channels [6].

In addition to their physiology and functionality, it is also worthwhile to understand the developmental origin of corneal nerves. These nerves develop through the migration and differentiation of neural crest cells from the trigeminal ganglion. Corneal nerve development studies in chicks and mice have shown that corneal nerves derive from neural crest cells within the ophthalmic lobe of the trigeminal ganglion [7]. Neural crest-derived cells migrate to the periocular mesenchyme, and this process is regulated by neuropilin (Npn)-1 and transmembrane roundabout (ROBO) in chicks. Studies have further shown that the receptor kinase Eph receptor A2 provides axon guidance to the migrating corneal nerves in chick embryos. Corneal nerves undergo negative regulation to develop the pericorneal nerve ring (allowing for nerve fascicles to accumulate and extend around the cornea), and positive regulation follows to enable corneal nerves to enter the stroma [7].

## 3. Cornea Nociceptor Receptors, Channels, and Neurotransmitters

### 3.1. Nociceptors and the Cornea

The corneal sensory nerve fibers are heterogeneous and include low mechanoreceptors, mechano and polymodal nociceptors, and cold thermoreceptors [5]. Nociceptor expression in the cornea is crucial for the sensation of noxious stimuli. Nociceptive fibers densely innervate the cornea; these nerve endings in the anterior corneal epithelium allow for the detection of noxious stimuli or threats to the eye [7]. Corneal nociceptors also play a critical role in initiating reflexes, such as blinking and tearing. The cornea contains many different types of nociceptors that respond to various stimuli based on the expression of different nociceptive channels, transducing painful stimuli via nociceptive fibers to the brain. After exposure to repeated noxious stimulation or injury, ocular polymodal nociceptors become sensitized, leading to pain. Sensory nerve endings, particularly in the corneal epithelium, undergo continuous remodeling. When these sensory nerves are damaged, they may undergo long-lasting changes to their excitability, leading to neuropathic pain [5].

The peripheral ends of these corneal nociceptive fibers express receptors and ion channels, enabling them to detect and respond to environmental changes associated with tissue injury, inflammation, or infection. Research in recent decades has implicated the TRP channel family as an essential target for the physiology and pathology of corneal sensation [8]. This diverse family of ion channels is found in various cell types throughout the body, where they play essential roles in sensory perception, cellular signaling, and homeostatic regulation.

### 3.2. Cornea Nociceptor Receptors and Channels

#### 3.2.1. TRP Channel Superfamily

The TRP channel superfamily comprises multiple subfamilies with distinct properties and functions. The TRP channel variants expressed in the human cornea include TRPA1, TRPV1-4, TRPC4, and TRPM8 [9,10]. TRPV1-4 and TRPM8 are involved in the perception of heat-induced and cold-induced pain, respectively. Studies of expression patterns in mouse corneas revealed that TRPV1, TRPA1, and TRPM8 are expressed on corneal axon terminals, with TRPV1 and TRPM8 expression on distinct populations of small-diameter C fibers and TRPA1 expression on specific intracellular axonal vesicles [6,11]. TRPV1, TRPC4, and TRPM8 are expressed in corneal epithelial cell membranes [6].

The TRPV (vanilloid) subfamily, which is the most studied, includes the TRPV1 to TRPV6 channels. TRPV channels are tetrameric proteins with six helical transmembrane domains [12]. TRPV1, in particular, opens in response to heat changes and transmits thermal pain and discomfort. There are several known agonists of TRPV1, the most well-known being capsaicin. Cryo-electron microscopy analysis of capsaicin-bound TRPV1 structures revealed that capsaicin binds to specific residues (Y512, S513, T551, E571), which disrupt the amino acid interactions that close the TRPV1 channel, leaving it open and active [12]. TRPV1 channels have also been shown to be acid-sensitive. Nociceptive neurons from mice without TRPV1 show significantly reduced proton sensitivity, and TRPV1 activation occurs at a pH threshold between 5.9 and 6.4, varying with temperature [13,14]. TRPV4 channels have been studied in a wide variety of neurons and glial cells in the nervous system. TRPV4 has been found in the CNS in hippocampal neurons, astrocytes, microglia, and oligodendrocytes. In the PNS, TRPV4 has been found in dorsal root ganglion (DRG) neurons, satellite glial cells, and Müller glial cells [15]. TRPV4 has also been found in the nerves of the cornea [16] and in human corneal epithelial tissues [16,17]. In the corneal epithelium, TRPV4 is intrinsically implicated in the photo-response to UVB radiation via acting in parallel with OPN5 signaling to promote Ca2+ signaling, leading to the release of pro-inflammatory and nociceptive cytokines and chemokines [18]. More specifically, TRPV4 knockout mice studies have demonstrated that UVB radiation leads to extracellular calcium influx through TRPV4 channels, and TRPV4 has been found to be increased in patients with acute sunburn and UV overexposure, suggesting that TRPV4 has a role in the acute response to UV radiation (pain and inflammation) [19]. Application of the TRPV4 inhibitor GSK205 to wild-type mouse paw pads demonstrated more sustained attenuation of thermally evoked nociceptive behavior in comparison to control mice, further suggesting that TRPV4 is a key modulator of UV-evoked corneal nociception [19]. Although these mouse studies implicate TRPV4 in corneal nociception, more studies in human corneal epithelial cells are warranted to elucidate the specific role that TRPV4 plays in human corneal nociception.

Another relevant TRP subfamily is TRPM (melastatin), which includes channels TRPM1 to TRPM8. These channels participate in thermosensation, ion homeostasis, and cellular signaling. Cold temperatures and menthol activate TRPM8, which produces a cooling sensation. TRPM2 is involved in the response to oxidative stress and calcium signaling. The TRPM4 and TRPM5 channels participate in ion homeostasis and taste perception, respectively. The distributions of cornea TRP channels also relate to their function. In the guinea pig cornea, Alamri et al. [20] showed that electrical recording activity registering in a specific cornea site in Bowman’s membrane comes from only TRPM8-expressing fibers, which form a single nerve terminal cluster in the corneal epithelium (Figure 3).

Other TRP subfamilies include TRPA (ankyrin), TRPC (canonical), TRPP (polycystin), and TRPML (mucolipin). Chemical irritants activate TRPA1 channels and contribute to nociceptive signaling and inflammatory responses. TRPC channels are involved in receptor-mediated calcium signaling and cellular excitability, while TRPP channels contribute to ciliary function and kidney development. TRPML channels are involved in lysosomal calcium release and membrane trafficking processes [8].

TRPA1 is implicated in acute pain due to various irritants and plays an essential role in corneal neovascularization. Vascular endothelial growth factor (VEGF) transactivates TRPA1 to control neovascularization and macrophage infiltration [8]. In terms of nociception, TRPA1 has been shown to be associated with components of pungent substances (cinnamon oil, mustard oil, etc.) that cause burning sensations in humans, including cinnamaldehyde, allyl isothiocyanate, and others [21]. Bradykinin, a well-known pain modulator, has also been shown to activate TRPA1 in CHO cells via a G protein-coupled receptor mechanism involving phospholipase C [21]. TRPA1 in humans, similarly to TRPM8, can also be activated by noxious cold temperatures, although these channels have distinct physiological functions, which will be described later [21].

TRPC4 channels are involved in corneal epithelial cell proliferation, allowing for the optimal response to epidermal growth factor (EGF) [8]. In general, TRPC channels are tetrameric proteins with four-fold symmetry and six transmembrane alpha-helices, and they form ion channels activated by stretch [22]. They are all mediated by calmodulin (CaM) as they all have CaM-binding sites on their cytoplasmic domains [23]. Notably, many TRPC family channels generally transduce mechanosensitive stimuli, although it remains unclear how mechanical stimuli activate these channels [8,23]. For example, a study involving the downregulation of TRPC1 by shRNA in mice DRG neurons in vivo showed a reduction in the number of neurons with stretch-activated responses, suggesting that TRPC1 has some involvement in mechanosensitive responses [24,25]. However, TRPC4, the specific variety found in corneal epithelial cells, is primarily involved in corneal epithelial regeneration through EGF activation of a G protein-coupled receptor mechanism involving phospholipase C (PLC) and inositol triphosphate (IP3), leading to downstream MAP kinase activity, triggering cell migration and proliferation [6]. Therefore, although other TRPC channels, such as TRPC1, have some involvement in detecting noxious mechanical stimuli, TRPC4 may not play a major role in corneal nociception.

#### 3.2.2. Acid-Sensing Ion Channels (ASICs)

Another essential group of ion channels involved in corneal nociception is the acid-sensing ion channels (ASICs). ASICs are associated with synaptic plasticity, learning, pain perception, and mechanosensation [26], and their activation leads to depolarization. Their role in nociception is the detection of tissue acidosis, which is commonly seen in inflammation and tissue damage [27]. ASICs respond to pH changes near the physiological range (6–7.4) [14]. Although ASICs and TRPV1 can both respond to acidic conditions, ASICs have a much more widespread distribution than TRPV1, with the majority of DRG neurons expressing ASIC1, ASIC2, and ASIC3 [13]. ASIC1a and ASIC3 have been found in whole-cell voltage clamp recordings of corneal afferent neurons [14]. Furthermore, application of ASIC3 agonists and antagonists modulates nocifensive responses to acid in rats, with agonists increasing the number of nocifensive responses (blinking, scratching) and antagonists decreasing the number of nocifensive responses in comparison to the control group [14]. ASICs are a major mediator of acid-induced corneal nociception.

#### 3.2.3. Mechanosensing Ion Channels

PIEZO2 channels are also crucial for corneal nociception, as they are one of the primary sensors of mechanical stimuli. These channels are large membrane proteins characterized by a central ion-conducting pore module and three peripheral mechanosensing blades containing 38 transmembrane domains. PIEZO2 channels have been identified in DRG and trigeminal ganglion neurons, including corneal nerves. Research suggests that PIEZO2 is associated with mechanical allodynia induced by inflammation or nerve injury [28,29]. PIEZO2 has also been shown to be important in corneal nerve mechanosensitivity, as PIEZO2 knockout mice exhibit a significant reduction in the percentage of corneal nerves that respond to mechanical indentation. More specifically, PIEZO2-dependent mechanosensitivity is present in both pure mechanonociceptor neurons and polymodal nociceptor neurons in the cornea, suggesting that PIEZO2 mechanosensitivity has some influence on all these nociceptive corneal neuron types [30].

#### 3.2.4. Coding Channels

Ion channels that conduct ions other than calcium are also involved in nociceptive neuron function. Two examples, Nav1.8 and Nav1.9, are voltage-gated sodium channels that exhibit depolarized voltage dependence of inactivation, making them more readily activatable than other Nav channels [31]. Nav1.8 specifically recovers rapidly from inactivation, producing more than 60% of the current underlying the depolarization phase of action potentials [32]. Nav1.8 and Nav1.9, therefore, are important for repetitive action potential firing, modulating nociceptor excitability, which is why therapeutic treatments target these channels for pain management [31,32].

Potassium channels are also involved in nociception. Modulation of potassium currents allows for the modulation of neuronal excitability. Specifically, axons involved in nociception express specific combinations of Kv1 channels, including Kv1.1 and Kv1.2, and nerve injury leads to reductions of Kv currents, increasing nociceptor excitability and hyperalgesia [33]. Two-pore-domain potassium (K2P) channels also contribute to nociceptive signaling by regulating the resting membrane potential. Channels like TRESK, TREK 1, and TASK are K2P channels expressed in nociceptive neurons. These channels maintain neuronal excitability and modulate pain sensitivity to mechanical, chemical, and thermal stimuli [34].

#### 3.2.5. Interactions and Distinctions Between Nociceptive Channels

Nociceptive channels do not act in isolation. For example, TRPC1 and TRPC6 (mechanosensitive stretch-activated channels) have been shown to work together with TRPV4 to mediate hyperalgesia to mechanical stimuli and nociceptor sensitization. Injection of selective stretch-activated ion channel inhibitors reverses the mechanical hyperalgesia caused by inflammatory mediator injection, and antisense oligodeoxynucleotides against TRPC6 reverse the mechanical hyperalgesia caused by thermal injuries and TRPV4-selective agonists [35].

TRPA1 and TRPV1 interact by forming a complex on the plasma membrane of neurons, allowing TRPV1 to modulate TRPA1 by controlling its voltage–current relationships, independent of intracellular calcium. Single-channel analysis of CHO cells with TRPA1 or TRPA1-TRPV1 coexpression showed that the coexpression of TRPV1 and TRPA1 causes significant FRET interactions between the two channels, and the combined TRPA1-TRPV1 complex produces a greater open probability of TRPA1 in the presence of mustard oil in comparison to cells with TRPA1 alone [36]. The TRPA1-TRPV1 complex enables sensory fibers to detect noxious heat (TRPV1) and noxious cold temperatures (TRPA1), with TRPV1 modulating TRPA1 activity.

TRPV1 and ASICs are both found on corneal afferent neurons and can both respond to acidity, but their properties are distinct. TRPV1 produces large inward ion currents at very acidic pH (5 or 4), while ASICs mediate smaller transient currents at pH 5 or 6 [13]. Despite these distinct pH ranges, both channels commonly contribute to the increased sensitivity of isolectin B4 (IB4)-negative unmyelinated DRG neurons compared with IB4-positive neurons in mice [13].

Although TRPA1 and TRPM8 respond to cool temperatures, they have distinct physiological functions. TRPM8 can be activated by menthol or moderately cold temperatures (28 °C), with increasing ionic currents down to 8 °C. In contrast, TRPA1 can be activated by noxious chemicals (cinnamaldehyde, mustard oil, etc.) or cold temperatures at an average threshold of 17 °C [37,38]. While TRPM8 does contribute to the sensation of noxious cold as well as non-noxious cool temperatures, TRPA1 appears to have a less direct role. Most TRPA1-expressing rat DRG neurons were found to have little to no response to cold temperatures. Still, in the small group of TRPA1-expressing neurons that did respond to cold, a cold-activated background calcium influx was observed that could be modulated by TRPA1 antagonists, which reduced the influx [39,40]. Furthermore, TRPA1 knockout mice show only partial deficits in cold sensation [41]. This suggests that TRPA1 may contribute to cold hypersensitivity: increased TRPA1 activity can bolster the response to noxious cold temperatures [42]. TRPA1, therefore, is not a key contributor to the sensation of cold itself but rather a modulator of that sensation. Notably, not all cold-sensing neurons express TRPM8, and the presence of TRPA1 cannot explain these neurons’ cold-sensing ability, given that a third of cold-sensitive mouse DRG neurons had neither TRPM8 nor TRPA1 [43]. A second independent calcium influx mechanism must be responsible for cold transduction in these TRPM8-deficient neurons.

Finally, it should be noted that most studies of TRP channels have been conducted on rodent corneas or neurons, and species-specific differences in TRP channel expression are possible. Studies in human corneal tissues will further clarify the specific roles that these TRP channels play in corneal nociception.

#### 3.2.6. Age- and Sex-Related Variation in Ion Channel Expression and Function

In general, nociceptive TRP channels lose functionality with age. A study analyzing TRPV1 and TRPV2 expression on the trigeminal ganglion neurons of young and aged mice found that young mice have a higher expression of TRPV1 small-diameter neurons and a lower heat head withdrawal threshold, indicating that aging is associated with a decline in TRPV1 expression and a decreased sensitivity to heat stimulation [44]. Similarly, a study analyzing TRPA1 responses to cold temperatures in young and aged mice found that TRPA1 activity declines with age [45]. TRPM8-expressing corneal cold nociceptor neurons in aged mice have lower background activity and abnormal responsiveness to cooling pulses compared with those of 3-month-old mice [46]. Interestingly, a study showed that TRPM8 morphological and functional changes in aging occur in parallel with altered basal tearing, suggesting that a decline in TRPM8 function is linked to poor maintenance of the tear film, explaining the higher prevalence of dry eye disease (DED) in elderly patients [46]. Overall, the decline of TRP channels with age underscores the need to consider their physiological function when analyzing corneal nociceptive pathology and pharmacology.

Differences in nociceptive ion channel function between males and females must also be considered. Estrogen has been shown to upregulate TRPV1 expression through both a classical pathway and a non-classical pathway. The classical pathway involves estrogen binding to its typical nuclear estrogen receptors in DRG neurons, where it binds to estrogen response elements located in the TRPV1 gene promoter [47]. The non-classical pathway involves estrogen activation of G protein-coupled receptor 30, which triggers an intracellular signaling pathway that activates protein kinase C, which phosphorylates and activates TRPV1 [48]. Through these pathways, estrogen contributes to the greater TRPV1 sensitivity observed in females compared with males [49]. However, estrogen can also inhibit P2X purinoceptor 3 (P2 × 3), leading to an anti-nociceptive effect as well; thus, estrogen is not entirely pro-nociceptive [49]. TRPM8 also expresses some sexual dimorphism, with testosterone–TRPM8 interaction in males leading to the recovery of mechanical sensitization in mice with chronic migraine [50]. Further inquiry into the sexual dimorphism of nociceptive channels specific to the cornea is required to identify sexual differences in human corneal nociception.

### 3.3. Corneal Neurotransmitters

Sensory corneal nerves, including nociceptive corneal nerves, rely on specific neurotransmitters such as substance P (SP) and calcitonin gene-related peptide (CGRP) to trigger action potentials in response to sensory stimuli [51]. Sympathetic corneal nerves express norepinephrine, serotonin, and neuropeptide Y to conduct action potentials in response to autonomic control. Other corneal nerve transmitters have been detected, including brain natriuretic peptide, vasopressin, neurotensin, and beta-endorphin [51]. To date, 18 unique neuropeptides and neurotransmitters have been detected in corneal nerves, and there are still many with unknown nerves of origin (Table 1).

## 4. Corneal Nerve Pathology

The cornea, which is innervated by many sensory nerve fibers, detects environmental stimuli and transmits signals to the CNS. Thus, corneal nerve density, morphology, and function alterations can cause ocular changes and discomfort. Because the corneal nerves are necessary for blinking and tearing reflexes, symptoms such as dryness, eye discomfort, pain, fatigue, lacrimation, burning, itching, and photophobia can manifest in ocular pathologies involving corneal nerve damage [53].

### 4.1. Corneal Nerve Lesions and Inflammation

Corneal sensory nerves send afferent signals to nuclei in the brainstem, which then return various efferent signals. For instance, autonomic nerves related to cranial nerve VII (the facial nerve) drive tear secretion in response to corneal nerve stimulation. Injury to the cornea can disrupt corneal nerves, leading to dysfunction of these reflexive neural circuits. Corneal nerves also produce many neurotrophic factors that help support corneal nerve growth, corneal epithelial growth, and regeneration [7]. Destruction of corneal nerves can produce neurotrophic keratopathy directly (via loss of supportive neurotrophic factors, leading to poor corneal nerve and corneal epithelial cell maintenance and regeneration) or indirectly (via dysfunction of neural circuits involving the corneal nerves) [7]. For example, in DED, corneal nerve lesions can disrupt reflective tear production, leading to deterioration of the tear film and ocular dryness. Corneal nerve injury also results in inflammation, and the release of inflammatory mediators can lower the threshold potentials of nociceptive nerve endings, intensifying pain signaling and leading to peripheral sensitization [54]. Over time, peripheral sensitization of the cornea can lead to central sensitization, with central sensory neurons becoming more sensitive to pain, leading to increased pain awareness [54]. Notably, a separate set of nociceptors exists that is activated only by local inflammation, as inflammatory cells can also trigger neuropathic pain [7]. Corneal nerves respond to inflammation by releasing neurotrophic factors that assist in the maintenance of the cornea and wound repair [7]. As examples, SP and epidermal growth factor (EGF) are released from corneal C fibers following inflammation, and they stimulate corneal epithelial proliferation and healing [55]. Nerve growth factor (NGF), brain-derived neurotrophic factor (BDNF), and neurotrophin-3 (NT-3) also have similar functions. Long-term corneal nerve dysfunction can impair the release of neurotrophic factors, leading to further disruption of corneal nerve and epithelium maintenance.

### 4.2. Dry Eye Disease

In DED, the tear film is disrupted, which causes ocular discomfort, visual disturbances, and possible damage to the ocular surface. The chronic dryness and inflammation associated with the condition are thought to lead to degeneration and remodeling of corneal nerves (Figure 4). Studies in patients with DED have shown reduced corneal nerve density and alterations in nerve morphology, including nerve tortuosity and beading [56,57]. In many patients with DED, sub-basal nerve density also decreases. One study found that increased sub-basal nerve density in DED correlates with better response to treatment [58]. Reduced sensory input from the ocular surface due to dry eye-induced corneal nerve loss, which can also be induced by conditions such as aging and refractive surgery, can hinder reflexive tear secretion and increase tear loss due to evaporation, increasing dryness and further exacerbating corneal nerve damage [56]. Even as corneal damage occurs with dryness, corneal nerve regeneration also occurs. Evidence suggests that the corneal pain sensitivity experienced in DED may be due to the hyperexcitability of regenerated nerves [4].

Sensitization is believed to be mediated by inflammatory mediators such as cytokines that induce the phosphorylation of ligand-gated channels and modify voltage-gated channels [59]. TRPM8 channels are widely recognized to be overactivated in dry eye-related conditions, leading to ocular pain and cold allodynia [60]. Additionally, crosstalk occurs between TRPM8 and TRPV1, which enhances the activation of TRPM8-expressing neurons. TRPV1 expression, and particularly its overactivation, has also been found to be required for axonal degradation in DED [61]. Huang et al. further found that TRPV1 contributes to itching in dry eye conditions through a histaminergic pathway, while TRPA1, which is frequently co-localized with TRPV1, contributes to itching through a histamine-independent pathway [62].

**Figure 4 ijms-26-04663-f004:**
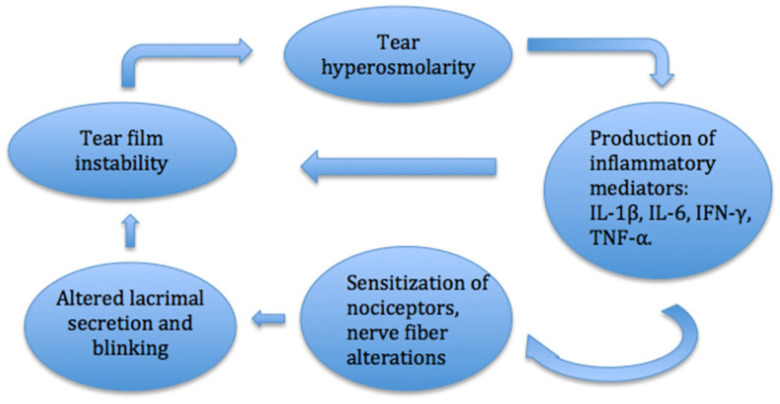
Inflammatory pathway of DED. The inflammatory pathway in the cornea leads to changes in the levels of cytokine mediators and nociception, which can disrupt the ocular surface and cause dry eye symptoms. Modified from Mikalauskiene et al., 2021 [63].

### 4.3. Refractive Surgery

Laser eye surgery procedures such as laser-assisted in situ keratomileusis (LASIK), photorefractive keratectomy (PRK), and small incision lenticule extraction (SMILE) correct refractive errors to eliminate the need for contact lenses and glasses. During LASIK, a flap is created in the corneal epithelium before a laser ablates the stroma as desired. In contrast to LASIK, PRK involves complete removal of the corneal epithelium and reshaping the stroma with a laser to correct vision. SMILE is a flapless vision-corrective surgery in which a lenticule of corneal tissue is extracted from a small incision. Studies have demonstrated reductions in corneal nerve density and alterations in nerve morphology after laser eye surgery, which may contribute to symptoms such as dryness, reduced corneal sensitivity, and ocular discomfort (see Figure 5).

Laser in situ keratomileusis-induced neurotrophic epitheliopathy (LINE) is a complication of LASIK characterized by punctate epithelial erosions, worsening vision, visual fluctuation, and night vision problems [53]. LASIK surgery results in damage to corneal nerves, with an up to 80–90% reduction in nerve density compared to the preoperative level [65]. Studies have shown a complete absence of basal/epithelial and sup-epithelial nerves in the flap area shortly after surgery, with nerve deficiency persisting for years postoperatively [66]. Erie et al. found that sub-basal nerve density is still reduced by 51%, 35%, and 34% after 1, 2, and 3 years, respectively [67]. Stachs et al. also observed abnormal nerve morphology with thin, non-branching nerves and no whorl-shaped sub-basal nerve plexus present 2 years after LASIK [68]. Garcia-Gonzalez et al. found that corneal nerves had not fully recovered even 10 years after LASIK [69]. This nerve damage correlates with corneal hypoesthesia, which can persist for years after surgery [70]. The damage interrupts the nerve circuit to the lacrimal gland, which affects tear production and blink rate and causes tear film hyperosmolarity and inflammation [71]. Corneal damage leads to deficiencies in trophic factors, altering healing and neurotropism [53]. Additionally, nerve-mediated feedback from the cornea to the lacrimal glands modulated the expression of nerve growth factors and cytokines essential for homeostasis and epithelial wound healing processes [71]. Furthermore, in rare cases, the damage that LASIK imparts to corneal nerves can induce long-term corneal pain. A study by Mshirfar et al. found that 1 in 900 LASIK patients had developed neuropathic corneal pain by 9.6 months after surgery, with symptoms including pain, spontaneous burning, and light sensitivity [72]. Thus, the evidence is clear that LASIK damages corneal nerves and can impart long-term corneal pain due to damage to afferent nociceptive corneal nerve fibers (Figure 6) [73].

PRK also leads to nerve loss due to several factors. During epithelial removal, intra-epithelial nerve fibers and the sub-basal plexus are injured. Additionally, laser ablation damages corneal nerves in the anterior and middle stroma. This corneal nerve injury extends beyond the area affected by the laser ablation due to retrograde axonal death, resulting in nerve loss beyond the dimensions of the laser ablation [53]. Corneal sensitivity is impaired after PRK, with levels decreasing to approximately 75% of preoperative values [74]. Superior hypoesthesia and lower tear break-up time have been observed after LASIK compared with PRK, although both procedures result in reduced tear secretion without a significant difference [75]. Nerve degeneration begins post-procedure and can persist for up to 7 days. Studies have shown that the reduction in corneal nerves persists for up to 12 months after PRK, with some studies indicating decreases in nerve function and abnormal morphology of the sub-basal plexus for even longer times [53]. Erie et al. found that sub-basal nerve density was still 40% of the original value at 12 months and took 24–36 months to recover completely [67]. Other studies found that after PRK, some corneas still had reduced nerve fiber bundle density after 5 years [65,67,76]. Although corneal morphology and nerve distribution were comparable to those in control eyes at 5 years post-surgery, persistent reduction in nerve density and incomplete neural recovery have been reported in some cases [53]. Because PRK can lead to corneal nerve injury and degeneration, just as in LASIK, corneal nociception is heavily impacted. When corneal nerves are injured, postoperative pain can occur due to the release of pain-inducing inflammatory factors like prostaglandins and neuropeptides, leading to pain that usually subsides within 72 h when re-epithelialization is complete [77]. Notably, PRK-related pain is generally reported to be worse than LASIK-related corneal pain. A study comparing postoperative outcomes in eyes treated with PRK vs. thin-flap LASIK (one eye per procedure per patient) found significantly higher levels of pain for eyes treated with PRK compared with eyes treated with LASIK, with the pain lasting a longer time after PRK than after LASIK on average [78]. This is likely because PRK causes postoperative exposure of the stromal layer of the cornea, which is highly innervated, to cytokines and eyelid movement, resulting in intense discomfort, pain, and delayed visual rehabilitation [79]. In comparison with PRK, LASIK does not cause damage to the corneal epithelium; thus, it results in less pain and faster visual recovery [79]. Another study of PRK nociception concluded that PRK nociception is caused by increased excitability of corneal trigeminal ganglion neurons after PRK-related damage to peripheral axons in the cornea. They also found that nerves damaged by PRK had reduced responsiveness to mechanical and chemical stimuli due to the loss of stretch-activated and acid-sensitive channels like TRPV1 and ASICs [80]. Overall, PRK causes greater nerve damage and nerve loss than LASIK, leading to higher levels of postoperative pain and long-term loss of corneal sensitivity to certain stimuli.

SMILE does not disrupt the ocular surface as much as LASIK, resulting in minor neural remodeling and improved biological healing compared with other procedures like LASIK [73]. Studies have shown that SMILE is associated with a lower drop in corneal sensitivity than LASIK. Additionally, SMILE preserves more of the anterior stromal nerves in the cap. In contrast, all nerve fibers from the epithelium to the anterior stroma are transected in LASIK. SMILE is associated with only a 23% reduction in sub-basal corneal nerves compared to 75% with LASIK [81]. Further, tear film performance and ocular surface health are better after SMILE, as indicated by lower tear osmolarity, longer tear breakup time, and Schirmer tests after SMILE [82,83]. These results suggest that LASIK damages corneal nerves most due to its invasiveness and disruption of corneal architecture and is least damaged by SMILE due to its minimal invasiveness [84]. It is postulated that the inflammatory environment caused by refractive surgery leads to changes in nociceptive channel expression, such as TRPV1 upregulation, which causes postoperative dryness and discomfort [53]. Other channels, such as TRPA1 and TRPM8, are also likely involved, based on previous research implicating them in other inflammatory conditions [85]. However, more research is needed to elucidate the exact mechanisms and changes in the channels involved.

### 4.4. Keratitis

Corneal inflammatory conditions, such as infectious keratitis, can lead to inflammation and damage to the corneal nerves. Inflammatory mediators released during the immune response can directly affect corneal nerve function and integrity, leading to nerve density and morphological alterations. In one study, infectious keratitis patients who experienced severe loss of corneal nerves during acute infection showed increased corneal nerve density in the first 6 months after infection resolution. However, even with this nerve regeneration, nerve density never recovered completely [86]. Another study on neurotrophic keratitis (NK), characterized by damage to the trigeminal nerve, found evidence of reduced corneal sensitivity, breakdown of the corneal epithelium, and impaired healing [87]. The loss of corneal sensitivity in NK cannot be simply explained by trigeminal nerve damage. NK manifests as changes in the corneal epithelium, ranging from irregular, dry, and cloudy epithelium to superficial punctate keratopathy, epithelial defects, ulcers, stromal melting, and changes in keratocytes, which are more likely mechanisms for loss of corneal sensitivity [87,88]. Additionally, impaired corneal sensitivity affects tear film production and blink rate due to reduced trigeminal reflexes. Further, altered or reduced sensitivity to stimuli such as touch, temperature, and pain can occur due to direct corneal nerve damage [54]. Acosta et al. found that in anesthetized animals with induced ultraviolet keratitis, mechanonociceptor fibers had a reduced mechanical threshold and increased responsiveness; similarly, chemo-nociceptors also showed increased responsiveness to chemical stimulation [89]. Furthermore, cold thermoreceptors (TRPM8-related) had much lower activity and reduced responsiveness to cool pulses [89]. Despite increased sensitivity to mechanical and chemical noxious stimuli, decreased sensitivity to other forms of pain often occurs, implying differential effects of inflammation on different nociceptive fibers or nociceptive channels (TRPV1, TRPM8, etc.).

### 4.5. Corneal Nerve Growth and Regeneration

The corneal nerves themselves contain a variety of neurotrophic factors that are essential to sustaining the normal cornea and facilitating corneal wound healing [90]. These factors include SP, CGRP, EGF, NT-3, NGF, and BDNF [90]. In response to inflammatory conditions, these factors are released from unmyelinated C fibers in the cornea to maintain corneal health. More specifically, SP and EGF are released directly from C fibers when inflammation occurs to stimulate epithelial proliferation and wound healing of the cornea. This process directly implicates the nociceptive fibers in corneal regeneration and maintenance. Notably, any changes in the signaling of neurotrophic factors can impair corneal nerve function and development, including nociceptive corneal nerves. For example, blocking tyrosine receptor kinase, a receptor of NGF, causes reduced responses from nociceptive neurons, stromal nerves, and corneal epithelium upon exposure to noxious stimuli [7].

In addition to the corneal nerves, several different cell types secrete factors that modulate nerve growth, especially during nerve regeneration. Corneal epithelial cells participate in the growth/regeneration of corneal nerves through phagocytosis of axonal debris and secretion of multiple growth factors (NGF, glial cell-derived neurotrophic factor [GDNF], BDNF, NT-3, and NT-4) that support nerve growth [7,91]. Furthermore, keratinocytes in the corneal stroma are crucial for supporting corneal nerve growth and regeneration; any reduction in keratinocyte density slows such regeneration [92]. Inflammatory cell infiltration of the corneal stroma, especially after injury, has increased BDNF expression, leading to increased corneal nerve regeneration [93]. The corneal epithelium is crucial for corneal nerve growth, as it releases several neurotrophic factors (NGF, GDNF, BDNF, NT-3, NT-4) to promote axon growth and support nerve tropism. Likewise, as stated previously, the corneal nerves are crucial for the maintenance and wound healing of the cornea, implying a mutually beneficial relationship between the nociceptive corneal nerves and the non-nervous corneal tissues. Corneal nociception is clearly linked to corneal regeneration.

Nociceptors and nociceptive channels/receptors also play important roles in corneal nerve development and nerve regeneration after injury. Jiao et al. studied the role of TRPV1 in corneal nerve regeneration after incision of the central cornea and epithelium to induce nerve damage in mice [94]. In their study, mouse eyes were injured and allowed to heal for 1 week before staining with rabbit anti-TRPV1 antibody to identify the corneal nerves expressing TRPV1. Their study found that TRPV1+ corneal nerves were present at a higher nerve density compared with TRPV1− nerves [94]. This result indicates that TRPV1 promotes more extensive corneal nerve regeneration after nerve injury. They also found that corneal epithelial immune cells are closely associated with TRPV1+ nerves. In both intact and injured corneas, CD45+ dendritic cells in the epithelium were found to be associated with TRPV1+ nerve fibers but not with TRPV1− nerve fibers [94]. In addition, TRPV1 was co-expressed with CD45+ stromal leukocytes and macrophages. The TRPV1 nociceptive channel highlights the crucial role of nociceptive receptors in promoting corneal nerve regeneration. These channels are closely associated with immune and inflammatory responses and have key roles in the corneal injury response and nerve regeneration (Figure 7). The studies of these channels in the cornea may provide insights into their effect on other organs in the body that also express the same type of channels. While the cornea is most accessible for these studies and free of the vasculature, pinpointing the specific role of these channels could be more feasible and lead to a wider understanding of their effect on sensory neurons.

TRPC channels have also been implicated in nerve guidance during corneal nerve growth and regeneration. During nerve growth, a motile structure called a growth cone guides nerve growth in specific directions [96]. It is widely accepted that the growth cone direction is controlled by a gradient of extracellular guidance factors that produce a calcium ion gradient inside the cone. This gradient directs the cone to turn towards the side with the highest calcium concentration [96]. Because TRPC channels are permeable to calcium and prevalent in the nervous system, TRP channels likely play a role in directing nerve growth. Indeed, three studies independently determined that TRPC channels control the turning of the growth cone among cultured cerebellar granule cells of rats and embryonic spinal neurons of Xenopus [97,98,99]. These studies used gradients of chemoattractants like BDNF or netrin to affect the direction of growth cone turning. They found that both the turning and calcium ion elevations in the growth cone were disrupted upon inhibition or small interfering RNA (siRNA)-mediated knockout of TRPC channels. Subsequently, TRPC control of the growth cone was found to be dependent on phospholipase C, an enzyme involved in converting phosphatidylinositol bisphosphate (PIP2) into inositol trisphosphate and diacylglycerol. Upon activation, TRPC releases internal calcium ion stores in cells [96]. TRPC channels mediate nerve growth direction by responding to chemoattractants that open the TRPC channels to reduce calcium gradients. Overall, TRPC channels broadly promote and modulate corneal nerve growth and regeneration in the cornea and other organs.

In summary, corneal nociception in the context of nerve damage is highly dependent on regenerative factors to restore nociceptive function. TRP channels, known to be involved in corneal nociception, also play important roles in nerve regeneration outside the cornea, suggesting that these TRP channels may also influence corneal nociceptor regeneration.

## 5. Nociceptors, Inflammation, and Sensitization

The relationship between corneal nerve regeneration/growth and various neurotrophic/growth factors and nociceptive TRP channels has been emphasized. It is also important to consider the role of inflammation in corneal nociception and nociceptor regeneration. As previously discussed, corneal sensation is determined by nociceptors stemming from the sub-basal plexus, which respond to noxious stimuli that activate membrane TRP channels. Sensitization, which is defined as increased responsiveness and activation due to a reduced threshold in response to stimuli, can occur in corneal nociceptors exposed to corneal inflammation. Many conditions can trigger inflammation, including infections, injuries, chemical exposure, and DED. The intricate relationship between nociceptor growth and sensitization involves a complex interplay between inflammatory cells and cytokines [100]. Inflammatory mediators released by damaged tissue and immune cells such as ATP, H+, neurokinin A, tumor necrosis factor-alpha (TNF-α), prostaglandin E2 (PGE2), and inflammatory interleukins (interleukin IL-1β and IL-6) can lead to the opening or modification of ion channels, resulting in nerve excitation or increased sensitivity [100,101,102]. These substances fine-tune immune cell responses, shaping the inflammatory microenvironment of nociceptor neurons.

Immune cells, particularly those strategically located at peripheral nerve terminals, secrete various neurotrophins that modulate nociceptor neuron activity [103]. Notably, immune cells release NGF and BDNF, which maintain corneal nerve function and sensitivity, and when elevated in DED and inflammation, lead to nociceptor sensitization and pain [104]. Ro et al. found that higher levels of NGF increase neuronal sensitization and hyperalgesia, and low levels partially reverse nerve injury and heat-related hyperalgesia [105]. These effects of NGF are due to p38 MAPK activation in DRG cells, which increases TRPV1 in peripheral terminals and axon terminal sprouting [105]. NGF can also bind tropomyosin receptor kinase A (TrkA) on nociceptors, activating signaling pathways like PI3K/Src kinase. PI3K/Src kinase phosphorylates TRPV1, which rapidly embeds within the neuronal membrane [106]. Ultimately, there is a delicate balance of NGF activity at the injury site, which can increase local pain sensitivity and autonomy while promoting peripheral neuronal growth [107].

There are also direct interactions between nociceptive neurons and pro-inflammatory cytokines, whose interactions are pivotal in the modulation of pain perception. Pro-inflammatory cytokines directly engage nociceptive neurons, regulating central sensitization and hyperalgesia [108]. Several key cytokines are important in orchestrating cellular responses and tissue repair processes. Understanding the immune response and neuronal interactions after nerve injury is crucial for developing treatments for neuropathic pain and nerve regeneration. As previously discussed, EGF is a critical player in corneal epithelial wound healing as it regulates cell proliferation and migration following injury. IL-1 impacts corneal epithelial injury responses, inducing apoptosis in the corneal endothelium and affecting essential corneal growth mechanisms. TNF-α contributes to corneal inflammation modulation by inhibiting neovascularization and influencing corneal transparency, thus supporting regeneration processes. Transforming growth factor beta (TGF-β) counteracts EGF-induced corneal epithelial cell proliferation and migration to regulate cellular behavior during wound healing events [109]. These findings underscore the importance of cytokines in modulating nociceptor sensitization.

T cells, macrophages, and mast cells are also active players in nerve growth. Several microRNAs have been shown to play regulatory roles in neuronal function in settings of chronic pain [22]. Upon injury, DRG cell bodies release miR-21 packaged in exosomes [110]. Macrophages and monocytes ingest the exosome, and then miR-21 inhibits SPRY2 gene transcription within these cells. The inhibition of the SPRY2 protein allows for enhanced neurite growth from DRG neurons and axonal regeneration [111,112]. In a study observing the timing of immune cell infiltration into the site of sciatic nerve injury, T cells and macrophages appear and spread over the area within 2–4 days [113]. This process clears the path for nerve regeneration, as macrophages clear myelin debris containing myelin-associated glycoprotein, a neurite growth inhibitor. However, another study refuted the regenerative role of these immune cells based on results showing that T cells and macrophages remain at the site of injury several months after injury [114]. Macrophages release IL-6, among other cytokines, which triggers sympathetic nerve fiber terminal growth into “basket-like structures” surrounding large-diameter sensory neurons [114]. Both macrophages and satellite cells coordinate to release NGF and NT-3, which are drivers of sympathetic fiber basket development [114]. In the cornea, specific populations of immune cells are responsible for corneal nociceptor regeneration. Resident intraepithelial dendritic cells, macrophages in the anterior corneal stroma, and infiltrating immune cells (neutrophils, T cells) respond to corneal inflammation by releasing cytokines that support corneal nerve regeneration, such as VEGF [115]. Li et al. found that wild-type mice with depleted neutrophils had significantly less corneal nerve density, and injured wild-type mice treated with anti-VEGF antibodies had impaired corneal nerve regeneration [116].

Other immune cytokines may also influence corneal nerve regeneration. Macrophages and Schwann cells produce cytokines that alter the delicate balance in nerve growth. Post-injury, matrix metalloproteases disrupt the blood–nerve barrier and release neuropeptides such as CGRP, SP, bradykinin, and nitric oxide to promote hyperemia and swelling [114,117].

This process attracts monocytes and T lymphocytes to the injury site, while chemokines like CCL2 and CCL3 enhance inflammation. Additionally, cytokines such as IL-1β, IL-6, TNF, and leukemia inhibitory factor (LIF) further amplify inflammation. Neuregulin binds to Schwann cell receptors post-injury, triggering signaling pathways that lead to demyelination and Schwann cell proliferation. Schwann cells release neurotrophic factors like NGF and GDNF, prostaglandins, and cytokines to sensitize nociceptors and modulate sensory neuron gene expression. Purinergic P2 receptors in the DRG modulate neuronal and macrophage function, influencing nociceptor activity through pathways involving cytokines like IL-1β, IL-6, and LIF. Fractalkine-mediated signaling between neurons and glia also contributes to neuropathic pain development [118].

More recently, Guerrero-Moreno and colleagues delved into the relationship between nociceptors, nerve growth, immune cells, inflammation, and cytokines in corneal regeneration [2]. Their study employed various experimental approaches, including calcium imaging on FM 1–43-labeled corneal neurons and custom-made chambers, to explore the role of nociception in encoding noxious stimuli and the development of pain-related sensory abnormalities due to nerve abnormalities. Their results demonstrated the bidirectional interaction between nociceptors and immune cells, as later emphasized by Maruyama [119], based on molecular and functional changes observed in trigeminal corneal neurons under various conditions. Specifically, they described alterations in neuropeptides like CGRP and SP, which contribute to peripheral sensitization and neurogenic inflammation. Furthermore, they reported activation of satellite glial cells and immune cells such as monocytes/macrophages in the trigeminal ganglion after corneal damage, suggesting their significant roles in modulating ocular pain.

## 6. Current FDA-Approved Nociceptor-Related Therapies for Ocular Pain

The importance of corneal pain in the contexts of corneal disease/surgery, corneal nociceptive fiber regeneration, and immune responses illustrates the need for therapies that can reliably control corneal pain. The mainstays of ocular therapies for corneal nociceptive pain target either the CNS or peripheral nervous system (PNS) and include corticosteroids, non-steroidal anti-inflammatory drugs (NSAIDs), and acetaminophen [120]. These are non-specific, as they primarily work at the level of the CNS and have some effects on the PNS. Glucocorticoids have dose-dependent and use-dependent effects on corneal nociceptive pain that may lead to either improved or delayed nerve regeneration, in addition to more systemic side effects [7]. NSAIDs are COX inhibitors that reduce inflammation by inhibiting PGE2 release, which decreases nerve sensitization by acting at the peripheral nociceptor terminal and the dorsal horn [121]. The mechanism of action of acetaminophen in nociceptor desensitization is unclear, but several possible mechanisms include CNS targets. These agents are broad-acting and non-specific and thus have low efficiency while causing many side effects [122].

Standard-of-care therapeutics have been developed to address nociceptive pain more specifically and effectively, namely, bandage contact lenses (BCLs), topical anesthetics, and immunomodulators such as serum tears (Table 2).

BCLs provide a direct barrier and thus a specific approach to reducing the risk of chronic nociceptive pain. BCLs are placed in the pre-corneal space, where they protect nociceptors from environmental irritants and mechanical friction [54]. Thus, when used immediately after surgical procedures, they promote nerve regeneration [123]. The prosthetic replacement of the ocular surface ecosystem (PROSE) is one of the most effective BCLs [54]. The PROSE is a liquid-filled scleral lens that helps prevent many of the issues with other BCLs, namely, ineffective moisture retention on the ocular surface and increased risk of infection after long periods of wear. However, BCLs do not provide a singular solution for treating nociceptive pain once the pain is established. This is especially the case when nociceptor sensitization and hyperalgesia have developed such that placement of the BCLs on the cornea leads to irritation [54]. Further research is needed to minimize the risk of irritation and infection using BCLs.

Pathological signaling of nociceptors is often reduced upon application of topical anesthetics, while accompanying inflammation is addressed with immunomodulators or serum tears [124]. An example of a novel topical treatment for dry eye-associated pain that the FDA recently approved is perfluorohexyloctane (F6H8) eye drops, which are approved for treating DED. F6H8 function has been suggested not to block the activity of a nociceptor, but rather to alleviate dry eye pathology through other mechanisms. In DED, nociceptive pain is induced by various factors, including dysregulation of TRPM8 channels, which detect dryness; thus, patients present with decreased tearing [125]. When applied to the ocular surface, F6H8 increases tearing and blinking via a long-lasting cooling effect, which drives a reflex response mediated by the TRPM8 thermoreceptors [126]. No significant adverse effects have been noted compared with saline solution as a control treatment. The most common adverse events reported in the KALAHARI clinical trial were vitreous detachment (1.9%), allergic conjunctivitis (1.4%), blurred vision (1.4%), and increased lacrimation (1.4%) [127]. Based on the promising results for disease-specific treatment options like F6H8, further development of more options specifically targeting nociceptive sensitization and the neuron growth cascade is needed [122].

Optive Plus is used to treat DED via the action of L-carnitine to inhibit TRPV1 receptors in the cornea. In DED, inflammation begins with tear film hyperosmolarity. L-carnitine is an osmoprotectant that reduces the production of reactive oxygen species [128]. No significant adverse effects have been reported. Kaercher et al. reported that Optive Plus eye drops containing L-carnitine can reduce dry eye nociceptive pain after 4 weeks [129]. However, it should be noted that L-carnitine has no direct effects on nociceptive fibers or nociceptive channels; it is mainly an anti-inflammatory drug that reduces pain.

## 7. Future Therapies and Targets

Given the limitations of current therapeutics, many therapeutics that target corneal nociceptors, as well as nociceptors in other organs, and which could have implications for corneal nerves, are currently in development. Furthermore, the therapeutic potential of many other nociceptor-related targets has yet to be explored. Most target TRP channels or mediators upstream of those channels, while others target general factors or other pathways that influence corneal nociceptors.

### 7.1. Therapies in Development

Corneal analgesics that are currently in development or have been approved include several TRP antagonists, NGF/TrkA signaling inhibitors, opioid receptor agonists, and neurokinin-1 receptor antagonists (Table 3). Each analgesic has a different mechanism of action and thus may be used for different corneal pathologies, as discussed below. Future evaluation of these analgesics may also benefit from combining various analgesics together.

#### 7.1.1. TRP-Based Therapeutics

A promising approach to addressing corneal nociceptive pain involves Ca2+ influx, which can be induced by hyperosmolarity, excess heat, decreased pH that induces pain, or other stimuli such as capsaicin. SAF312 is a highly selective TRPV1 antagonist with a good safety profile, even at the highest dose available. SAF312 binds noncompetitively to TRPV1 in corneal and conjunctival cells and inhibits the Ca2+ influx induced by nerve injury [130]. The attenuation of TRPV1 after nerve injury has been shown to increase nerve regeneration, which implies that overactivation of TRPV1 during injury can hinder nerve repair in addition to its role in nociceptive signaling [131].

Another study investigating inflammatory eye pain in DED found that chronic topical use of capsazepine can alleviate symptoms. Capsazepine antagonizes TRPV1, TRPV4, and TRPM8 and suppresses Ca2+ influx [132,133]. Additionally, capsazepine plays an immune-modulating role by reducing macrophage and eosinophil infiltration and proinflammatory cytokine release in corneal nociceptors [134]. Repeated exposure to capsazepine reduces corneal polymodal responsiveness to heat, cold, and acidic stimulation [132]. Another strategy is a combined treatment with L-carnitine and capsazepine. L-carnitine is made by the liver and found in conjunctival epithelial cells and corneal keratocytes. It also reduces TRPV1 activation. L-carnitine combined with capsazepine was shown to suppress Ca2+ influx through TRPV1, producing a marked reduction in corneal discomfort and pain [135]. Ultimately, combination treatment with capsazepine and L-carnitine can potentially prevent neuron sensitization and pain in post-nerve injury settings.

Activation of TRPM8 also increases the expression of Tac 1, thus inducing corneal inflammation [136,137]. Fakih et al. reported that a TRPM8 blocker (M8-B) decreases inflammation in the cornea and prevents its spread to the ipsilateral trigeminal ganglion [136]. They observed marked decreases in IL-1β, IL-18, CCL2, prostaglandins, and CX3CR1 in the trigeminal ganglion, suggesting that M8-B can potentially prevent further sensitization of corneal nociceptors, which contributes to pain [136]. Another study of two other TRPM8 antagonists, DFL23693 and DFL234488, determined that the application of these antagonists in vivo results in significant antinociceptive effects in neuropathic and orofacial pain stemming from the trigeminal ganglion [138]. Given that the corneal nerves stem from the trigeminal ganglion, these results suggest that TRPM8 antagonists may be useful for antinociceptive effects in the cornea. Counterintuitively, TRPM8 agonists can also be used to treat ocular pain in DED specifically. A pilot study investigating cryosim-3 (C3), a topical TRPM8 agonist, found that regular application of topical C3 to the eyelids of 15 patients led to reported decreases in eye pain intensity and increases in quality of life [139]. The likely mechanism of this therapy in DED is that TRPM8 agonism increases basal tear secretion and alleviates dryness-associated pain. The activation of TRPM8, as briefly discussed, is associated with the stimulation of basal tear secretion. Thus, TRPM8 agonists may indirectly reduce dry eye pain by maintaining the tear film rather than directly disrupting corneal nociception [140]. Although the FDA-approved TRPM8 activator F6H8 has already been discussed, the aforementioned studies of novel TRPM8 agonists show that there is still potential for further development. Overall, TRPM8 agonism and antagonism have advantages and disadvantages. While TRPM8 antagonism can directly block nociception, it may also disrupt basal tear production and worsen DED. TRPM8 agonism can address this concern, but it does not have any direct effects on corneal nociception.

#### 7.1.2. NGF-Based Therapeutics

In an interventional case series, Bonini et al. found that administration of topical murine NGF eye drops results in complete resolution of a persistent corneal epithelial defect in patients with neurotrophic keratitis [141]. In the first few days of treatment, reported side effects included hyperemia and ocular and periocular pain. Still, patients reported improved corneal sensitivity and visual acuity at the conclusion of the study. NGF eye drops offer a promising therapy for restoring ocular surface integrity, and further investigation is warranted.

Another target that could be used to modulate NGF release is Thy-1 YFP-positive myeloid-derived suppressor cells (MDSCs). In mouse models, MDSCs have been seen entering the cornea and secreting NGF after annular keratectomy. As a result, MDSC activity can induce nociceptor growth, leading to increased sensitivity [142].

#### 7.1.3. TrkA-Based Therapeutics

Other novel pain therapeutics targeting the NGF pathway in the early stages of discovery or pre-clinical development include small molecule-based inhibitors targeting the NGF receptors TrkA and p75NTR, which are involved in nociceptor neuron growth and apoptosis. Monoclonal antibodies that bind and neutralize TrkA are also in development. TrkA is one of two NGF receptors with a higher binding affinity for mature NGF, activating neurotrophic signaling [143]. Small-molecule NGF/pro-NGF inhibitors are also under pre-clinical investigation for their ability to disrupt NGF/proNGF binding to TrkA and p75NTR in the context of osteoarthritis [144]. p75 neurotrophin receptor (p75NTR) is the second receptor on NGF. p75NTR has a preferential binding with proNGF and can induce neurotrophic and apoptotic signaling [145,146]. LEVI-04 is an injectable p75NTR fusion protein studied in a phase 1 clinical trial, although no results have been reported (Levicept NCT03227796, 2021). While still in the early developmental stages, these small molecule-based inhibitors may be of therapeutic interest for modulating NGF-induced sensitization of nociceptive signaling pathways.

New targets for corneal nociceptive pain may be found by reviewing studies of nociceptive pain in other body areas, such as the joints. A phase 2 clinical trial showed modest improvement in osteoarthritic pain with a single intraarticular injection of a TrkA inhibitor at 8 weeks [147]. In contrast, treatment with an oral TrkA inhibitor (100 mg, twice daily) failed to improve pain and function in osteoarthritis patients after 4 weeks (Figure 8 [147,148]). These findings show the potential of developing a therapeutic alternative that is specific to the nociceptive pain pathway. Further investigations regarding the application of such approaches to corneal nociceptive pain and the route of administration are needed.

#### 7.1.4. Novel Mu-Receptor Therapeutics

Opioids bind to peripheral opioid receptors called mu-opioid receptors (MORs) at the nociceptor terminal [121,149]. The specific mechanism linking MORs to the responsiveness of TRP-like nociceptors involves the descending inhibitory pathway and the dorsal horn. In this context, opioids act on presynaptic MORs to inhibit gamma-aminobutyric acid (GABA) release from inhibitory interneurons, leading to increased descending inhibitory signals from the rostral ventromedial medulla (RVM) and locus coeruleus (LC) to the dorsal horn. Additionally, opioids directly activate descending inhibition in the RVM. At the dorsal horn, activation of presynaptic MORs at nociceptor terminals inhibits the release of excitatory neurotransmitters. In contrast, activation of postsynaptic receptors on second-order projection neurons induces hyperpolarization and suppresses their excitability. The overall effect decreases nociceptive signaling from the dorsal horn to higher centers, impacting the responsiveness of TRP-like nociceptors.

D-Ala, N-Me-Phe, Gly-ol (DAMGO), a topical MOR ligand, induced a substantial reduction in the responsiveness of corneal polymodal nociceptors in mice with corneal inflammatory pain [23]. This finding contributes to the growing literature indicating that topical opioid agonists show promise as a possible therapeutic option for chronic ocular pain [150].

#### 7.1.5. Neurokinin 1 Receptor Antagonist Therapeutics

Topical aprepitant, a neurokinin 1 receptor antagonist, is another novel treatment for ocular surface pain that has shown promise as a therapeutic agent [120]. Aprepitant is normally indicated for the prevention of chemotherapy-induced nausea and vomiting; however, aprepitant was found to induce decreased pain sensitivity and upregulation of BDNF, a known enhancer of neurogenesis, within the trigeminal ganglion when compared with diclofenac in mouse models [120]. Additionally, aprepitant was found to cause a reduction of corneal nociception while having no effect on corneal sensitivity [120].

#### 7.1.6. Dual Enkephalinase Inhibitors (DENKIs)

DENKIs reduce corneal pain by inhibiting the catabolic enzymes neprilysin, aminopeptidase N, and leukotriene A 4 hydrolase, which blocks the degradation of enkephalins. Studies in mice have shown that the resulting higher extracellular enkephalin concentration causes enkephalins to bind to corneal opioidergic receptors, leading to pain alleviation in several corneal injuries (trauma, toxin exposure, inflammation); in other words, DENKIs function as indirect-acting opioid receptor agonists [151]. DENKIs have demonstrated antinociceptive and anti-inflammatory properties in mouse corneas. Several DENKIs are being investigated in clinical trials but have yet to be approved for clinical use [151].

#### 7.1.7. Nav Channel Inhibitors

Because Nav channels are important for modulating the excitability of nociceptive neurons, especially Nav1.8 and Nav1.9, several therapeutics are being developed that target these channels. One example is VX-548, a subtype-selective Nav1.8 inhibitor, that was shown to reduce postoperative pain in studies on abdominoplasty and bunionectomy; however, the resulting pain relief was only partial, indicating that further development is needed [152]. Furthermore, several non-psychotomimetic phytocannabinoids, including cannabidiol (CBD) and cannabinol (CBN), have shown promising results as analgesics in pain models, and research has shown that Nav channels are a key target of these cannabinoids [153,154,155]. Further in vivo studies must be conducted to determine the potential of cannabinoids as Nav-inhibiting analgesics and whether they have potential benefits for nociception in the cornea specifically.

### 7.2. Potential Future Targets

There are several mediators of corneal nociception for which no targeted pharmaceutical research has been reported, but they seem promising for future antinociceptive therapeutics for the cornea (Table 4).

#### 7.2.1. PIRT

Targeting TRPV1 regulators such as PIRT (phosphoinositide-interacting regulator of transient receptor potential channels) holds the potential to positively influence nerve regeneration by influencing pain perception, inflammation, and immune responses. PIRT is a key regulator of nerve function, particularly in nociceptive neurons. PIRT plays a crucial role in modulating TRPV1 activity. Research indicates that PIRT positively regulates TRPV1 channel activity by binding to TRPV1 and phosphoinositides like PIP2 [156]. Its expression in nociceptive neurons within the DRG suggests a specific role in peripheral sensory functions, highlighting its potential importance in nociception and sensory processing. Furthermore, PIRT’s association with TRPV1 and its impact on sensory neuron function suggest a significant role in modulating nociceptive responses and potentially influencing nerve regeneration processes [157]. Further research into the mechanisms underlying PIRT’s regulatory functions could provide valuable insights into its therapeutic potential for enhancing nerve repair outcomes.

In addition to PIRT, other regulators play a significant role in influencing nerve regeneration processes. For example, Carlin et al. demonstrated that a Tsc2 deletion in nociceptors has been shown to enhance axon regeneration after nerve injury by activating the rapamycin complex 1 (mTORC1) signaling [158]. This activation improves initial axon growth, although the elongation rate becomes similar to that in controls within a few days post-injury. In addition, genetic activation of mTORC1 in nociceptors can enhance axon growth within a short period after nerve injury, suggesting a potential mechanism for improving nerve regeneration. Notably, nociceptor deletion of Tsc2 induces pro-regenerative non-neuronal responses, particularly involving macrophages, which are crucial in promoting axon outgrowth and enhancing regeneration. These findings underscore the significance of targeting nociceptor regulators like PIRT and understanding their interactions with signaling pathways such as that of mTORC1, which will offer new insights for enhancing nerve regeneration strategies and potentially lead to innovative therapies for improving nerve repair outcomes.

PIRT’s role in corneal nerve growth and regeneration further highlights its significance in sensory neuron function. Corneal nerves play a crucial role in maintaining corneal health and sensitivity, with factors like NGF promoting subepithelial nerve regeneration after keratoplasty [159]. NGF stimulates corneal sensitivity and nerve regeneration, making it a potential treatment for conditions like neurotrophic keratitis. Additionally, interventions targeting neurotrophic support, such as Cenegermin, have been associated with sub-basal corneal nerve regeneration and lasting epithelial healing, indicating significant improvements in corneal nerve regeneration and overall corneal health [160]. PIRT’s influence on TRPV1 and its indirect impact on corneal nerve growth and regeneration sheds light on the complex mechanisms involved in maintaining corneal sensitivity and promoting nerve regeneration.

#### 7.2.2. ASICs

ASICs are characterized by their activation in response to acidic changes in the pH of the corneal extracellular environment [14]. ASICs are in corneal-free nerve endings and are implicated in ocular surface inflammation and pain [14]. Currently, there are no approved therapeutics that target these channels directly to aid in the management of corneal pain. Studies on the specific mechanisms of these channels have been conducted using peptide toxins to better understand the underlying physiology and their possible role in reducing acute pain via inhibition [161]. Specifically, preliminary data obtained from using PcTx1 spider toxin showed an inhibitory effect on rat ASIC1a by increasing the activation of the channel to more alkaline pH values, which was attributed to a steady-state desensitization-promoting effect [161]. An additional study examining the deletion of the ASIC1 gene in mice observed a decrease in the M1–M2 macrophage ratio, leading to pain relief and decreased nerve degeneration [162]. The evidence for the role of ASICs in the production of corneal nociception and the current lack of targeted treatments leave room for further in-depth research of these channels to support the development of new therapeutics.

#### 7.2.3. TRPA1

In addition to TRPV-based therapeutics, TRPA1 shows promise as a viable target for reducing corneal nociceptive pain. The TRPA1 channel plays a role in the transduction of inflammatory nerve signals. IL-4 and IL-13 receptors signal through TRPV1 and TRPA1, which increase intracellular calcium levels and lead to signaling [163]. Thus, blocking TRPA1 should reduce inflammation and, as a result, neuropathic pain and itch. In an in vivo rat model, using a selective TRPA1 antagonist resulted in a reversion of corneal nociceptive pain induced by the chemotherapeutic oxaliplatin [164]. In another study, the Sigma-1 receptor was targeted to reduce TRPA1 functionality [165]. The Sigma-1 receptor plays a role in the trafficking of functional TRPA1 to the plasma membrane. Administration of a Sigma-1 receptor antagonist decreased TRPA1 plasma expression and reduced neuropathic pain. Based on these results, direct and indirect TRPA1 antagonists may be beneficial for reducing corneal neuropathic symptoms, although more research is necessary.

#### 7.2.4. TRPV4

TRPV4 is another TRP channel that may be a potential target for modulating corneal nerve regeneration and growth. TRPV4 has been shown to be important for the regeneration of nerves after injury [15].

Feng et al. analyzed wild-type and TRPV4 knockout mice, both with sciatic nerve injury, and observed that the TRPV4 knockout mice had delayed functional recovery, indicating that the absence of TRPV4 delayed recovery of the injured nerve [15]. Furthermore, Western blot analysis of injured wild-type and TRPV4 knockout mice revealed that the lack of TRPV4 prevented demyelination processes, further inhibiting remyelination [12]. Blocking remyelination would prevent the recovery stage of sciatic nerve injury, meaning that the absence of TRPV4 prevents nerve regeneration [15]. Because TRPV4 is also expressed in the cornea, local activation of TRPV4 could be a potential therapeutic solution to promoting corneal nerve regeneration after injuries resulting from trauma or procedures like LASIK.

#### 7.2.5. PIEZO2

Another possible target for future therapeutics is PIEZO2 channels, which have been associated with mechanical allodynia after nerve injury [28]. There are currently no small-molecule inhibitors that are selective for PIEZO2 over PIEZO1. PIEZO1 is a similar channel with a broader expression profile involved in many non-nociceptive processes like vessel formation, mean corpuscular volume regulation, and the cell division of epithelial cells [165]. Inhibitors that target both PIEZO2 and PIEZO1 include ruthenium red, gadolinium, and GsMTx4 (Grammostola mechanotoxin 4). However, research has shown that TMEM120A, an enzyme similar to long-chain fatty acid elongase, selectively inhibits PIEZO2 over PIEZO1 with a concomitant increase in cellular lipids, including phosphatidic acid and lysophosphatidic acid [165]. Another study also found that intracellular application of phosphatidic acid or lysophosphatidic acid, or prolonged extracellular exposure, selectively inhibits PIEZO2 but not PIEZO1 activity [166]. Furthermore, inhibition of phospholipase D selectively inhibits PIEZO2 over PIEZO1, likely through modification of the intracellular lipid content. There are clearly lipid regulators and mediators involved in selective PIEZO2 inhibition that future research may exploit to produce PIEZO2-specific therapeutics [165].

#### 7.2.6. K2p K+ Channels: TRESK and TREK1

TRESK and TREK1, TWIK-related (Spinal Cord) K+ channels, are part of the K2p K+ channel family that helps maintain the resting potential in neurons [27]. While research is limited, these channels may be valuable targets in modulating nociceptor depolarization. One study demonstrated that increased expression of TRESK channels in mice with spinal cord injury resulted in quicker paralysis recovery and decreased TNF-α levels [167]. Both TREK1 and TRESK are predominantly found in the trigeminal ganglion and DRG neurons and have been implicated in increasing nociceptor excitability, triggering migraine pain. While current research is limited regarding the presence of TRESK and TREK1 on the corneal surface, genomic studies have shown overlap with TRPV1 expression, implying a role in ocular nociceptive and thermoreceptive neurons. Moreover, there is limited knowledge pertaining to TRESK and TREK1 activity in the cornea. However, their sensitivity to pH in the extracellular environment is notable. TREK1 is a proton sensor inhibited by decreasing pH levels, resulting in depolarization of the neuron cell membrane by blockage of K channels. Similarly, TRESK activity is altered by intra- and extracellular pH levels; lower levels inhibit TRESK activity, while increased pH enhances its activity. Uniquely, TRESK can be regulated via calcineurin-mediated phosphorylation and intracellular Ca2+ concentration, similar to TRP channels. Based on this activity, TRESK and TREK1 may act together to depolarize nociceptor membrane potential, facilitating neuron activation. Further research is needed to understand the therapeutic potential of these receptors for addressing corneal nociceptive pain.

## 8. Conclusions

This review summarizes the current evidence for the general role of corneal nerves in sensory function and the detection of environmental stimuli. Additionally, it discusses the role of nociceptors and their regulators in modulating corneal nerve regeneration post-injury. Nociceptors, particularly the unmyelinated C-fibers and lightly myelinated A-delta fibers, are densely expressed in the cornea and play a vital role in ocular health, detecting noxious stimuli and transmitting pain signals to the CNS. Members of the TRP channel family, including TRPV1, TRPM8, and TRPA1, have emerged as key targets for modulating nociceptor sensitivity and corneal nerve regeneration. TRPV1 is activated by noxious heat, capsaicin, and inflammatory mediators, while cold temperatures and menthol activate TRPM8. TRPA1 is involved in detecting oxidative stress and inflammatory agents. Targeting these channels with specific agonists or antagonists can potentially reduce nociceptor sensitization and promote nerve regeneration.

NGF, BDNF, and GDNF are clearly crucial in supporting corneal nerve growth and regeneration. These factors are released by various cell types, including corneal epithelial cells, keratocytes, and immune cells, in response to injury or inflammation. Modulating the expression or signaling pathways of these neurotrophic factors could enhance nerve regeneration and restore corneal innervation. Furthermore, immune cells and inflammatory mediators also significantly influence nociceptor activity and corneal nerve regeneration. IL-1β, TNF-α, and IL-6 can sensitize nociceptors and contribute to neurogenic inflammation. Conversely, anti-inflammatory agents such as adenosine and enkephalins can decrease nociceptor sensitization and promote nerve regeneration. TRP channels of nociceptors also play important roles in promoting and modulating nerve growth and regeneration. TRPV1, TRPV4, and TRPC have been implicated in nerve growth and/or differentiation in the cornea and other organs via knockout mice studies and in vitro nerve growth studies.

Future research should focus on developing targeted therapies that modulate nociceptor sensitivity and promote corneal nerve regeneration. Potential therapeutic strategies and avenues for research include developing specific agonists or antagonists for TRP channels, including TRPV1, TRPM8, and TRPA1, to modulate nociceptor activity, as well as exploring the use of neurotrophic factors or their signaling pathways, such as PIRT, to enhance nerve growth and regeneration. Additionally, combinations of therapeutic approaches, such as TRP channel modulators, neurotrophic factors, and anti-inflammatory agents, may provide synergistic effects. Advanced imaging techniques such as OCT and IVCM provide non-invasive means to assess corneal nerve structure and morphology, offering valuable insights into conditions like DED and post-surgical changes. Furthermore, most nociceptive channel experimental evidence is derived from animal models, which can hinder inferences about human corneal nociception. Future research should use these animal models as a basis for translational research with human models to elucidate human-specific mechanisms of corneal nociception. The emerging therapeutic data will facilitate further evaluation and discussion of relevant details regarding clinical limitations, side effects, barriers to translation, efficacy, delivery methods, trial phases, toxicities, age- and sex-related differences in treatment efficacy and design, and the dual role of TRP channels in nociception and regeneration. It is also important to recognize the increasing role of artificial intelligence (AI)-based research in corneal nociception imaging and therapeutic development; future research will greatly benefit from AI enhancement in data analysis and identification of novel drug targets. Interdisciplinary collaboration combining AI computational modeling with clinical and experimental ophthalmology will be crucial for expediting the development of personalized and predictive pain management strategies for corneal pathologies.

In conclusion, continued investigation into the mechanisms underlying corneal nerve function, degeneration, and regeneration is paramount for advancing therapeutic approaches. Bidirectional interactions between neurons and the immune system, alongside emerging treatments, demand further optimization and evaluation in more extensive clinical trials. Ultimately, a comprehensive understanding of these processes will pave the way for more effective management of corneal neuropathic pain and for strategies to promote corneal nerve repair and regeneration, thus enhancing overall ocular health and patient outcomes.

## Figures and Tables

**Figure 1 ijms-26-04663-f001:**
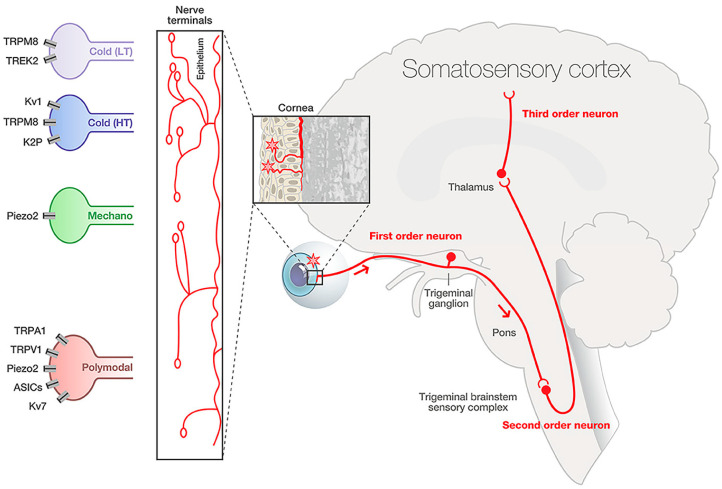
Corneal nerve pathway. The corneal nerve cell body is in the trigeminal ganglion. From here, axons innervate the cornea and synapse with second-order neurons in the trigeminal nucleus caudalis. These second-order neurons synapse on the thalamus, and third-order neurons synapse on the somatosensory cortex. Modified from Guerrero-Moreno, 2020 [2].

**Figure 2 ijms-26-04663-f002:**
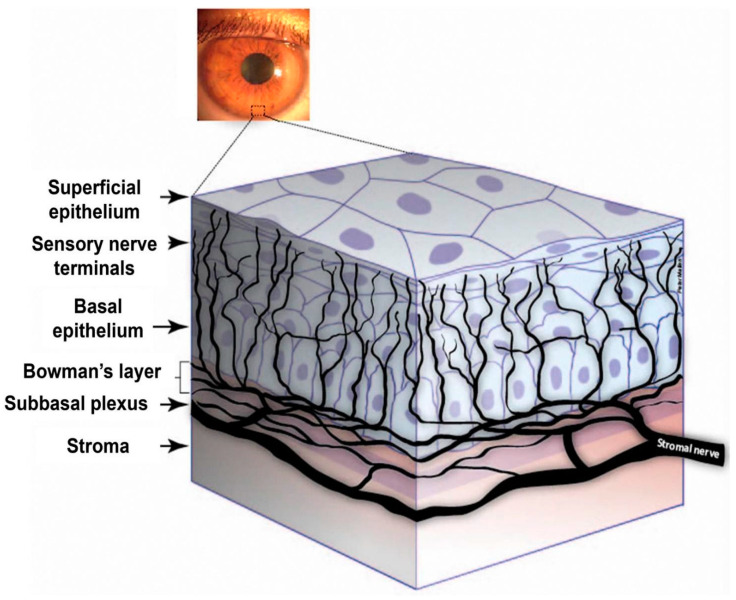
Diagram of human cornea innervation. The sub-basal nerve plexus is located between the Bowman’s layer and basal epithelial layer and contains unmyelinated nerve axons that project into the superficial epithelium, providing a high level of sensation. Modified from Cruzat et al., 2017 [4].

**Figure 3 ijms-26-04663-f003:**
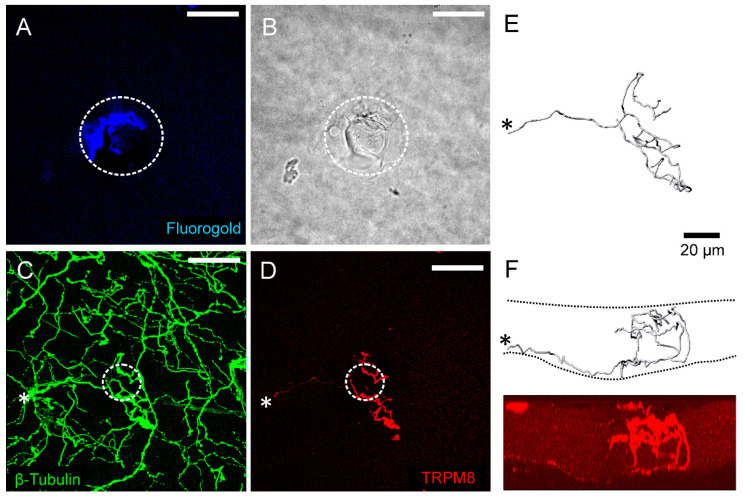
Analysis of a corneal recording site shows a unique expression of the TRPM8 ion channel. The three-dimensional reconstruction of the TRPM8 nerve terminal on a corneal nociceptor based on immuno-labeling shows that fibers expressing TRPM8 form a cluster in the corneal epithelium; they are derived from the same axon and respond equally to a stimulus. (**A**) Fluoro-gold staining in the white dotted circle shows the approximate location of the recording site. (**B**) Bright-field image of the same white dotted circle. (**C**) Axons in the recording site labeled by β-tubulin III-IR (green). (**D**) TRPM8-IR (red) nerve terminal within the recording site. The scale bar in (**A**–**D**) = 50 μm. Frontal (**E**) and side (**F**) views of 3D reconstruction of the TRPM8-IR nerve terminal. The asterisk marks the site where the parent axon of the TRPM8-IR nerve terminal enters the epithelium through Bowman’s membrane. Modified from Alamri et al., 2018 [20].

**Figure 5 ijms-26-04663-f005:**
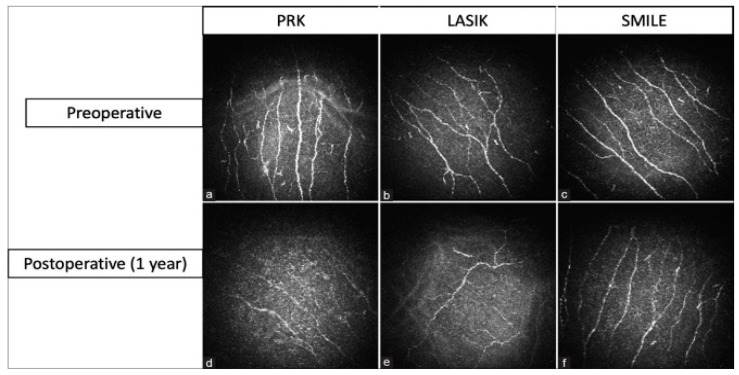
Confocal microscopy images showing corneal nerve fiber distribution before and after PRK, LASIK, and SMILE. Confocal microscopy images of preoperative and postoperative sub-basal corneal nerve fiber layer at 1 year after (**a**,**d**) PRK, (**b**,**e**) LASIK, (**c**,**f**) SMILE. Modified from Nair et al., 2023 [64].

**Figure 6 ijms-26-04663-f006:**
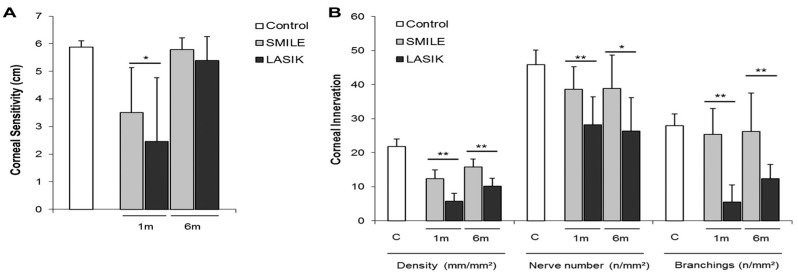
Corneal nerve sensitivity and innervation after SMILE or LASIK surgery. (**A**) LASIK results in a greater reduction in corneal sensitivity compared with SMILE. (**B**) LASIK leads to greater decreases in corneal nerve density, branching, and number compared with SMILE. (* *p* < 0.05; ** *p* < 0.01. Modified from Denoyer et al., 2015 [73].

**Figure 7 ijms-26-04663-f007:**
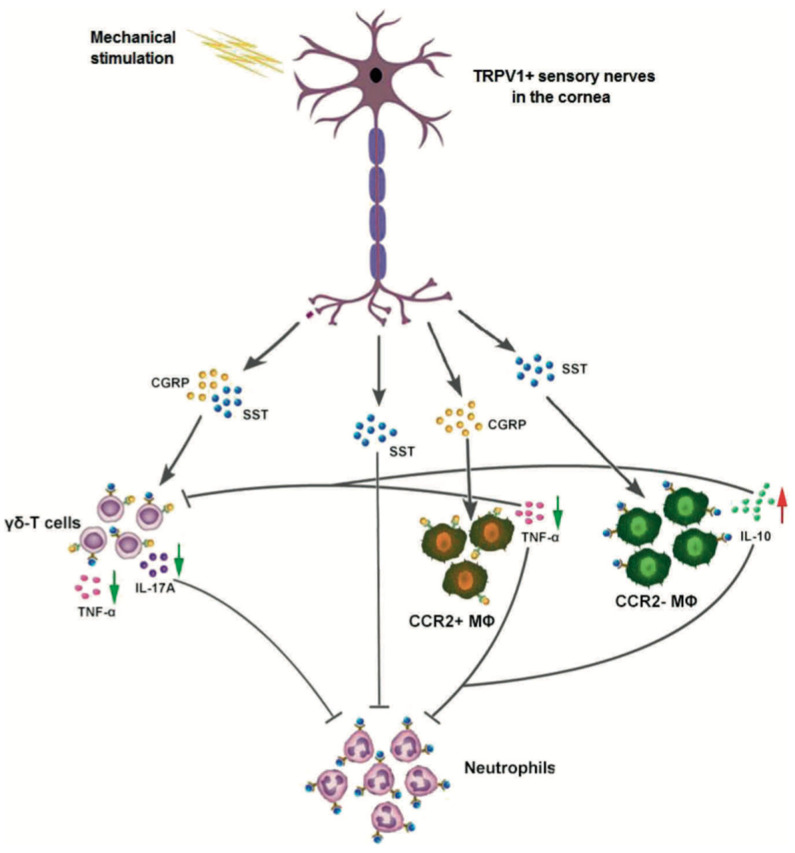
TRPV1 stimulation modulates corneal wound healing. Corneal abrasion activates TRPV1^+^ sensory nerve endings in the cornea to release the neuropeptides SST and CGRP, which reduce the degree of inflammatory response to corneal injury by binding to corresponding receptors expressed in different preferential ways on each of the three immune cell types. For neutrophils, the SST released from TRPV1^+^ nerve terminals acts directly on neutrophils via the SST-SSTR5 axis to inhibit their recruitment to the injured cornea. As for γδ T cells, both the CGRP and SST released from TRPV1^+^ nerve endings inhibited their infiltration and expression of the pro-inflammatory cytokines TNF-α and IL-17A via both the CGRP-RAMP1 and SST-SSTR5 pathways, thereby indirectly dampening neutrophil recruitment. Modified from Liu et al., 2022 [95].

**Figure 8 ijms-26-04663-f008:**
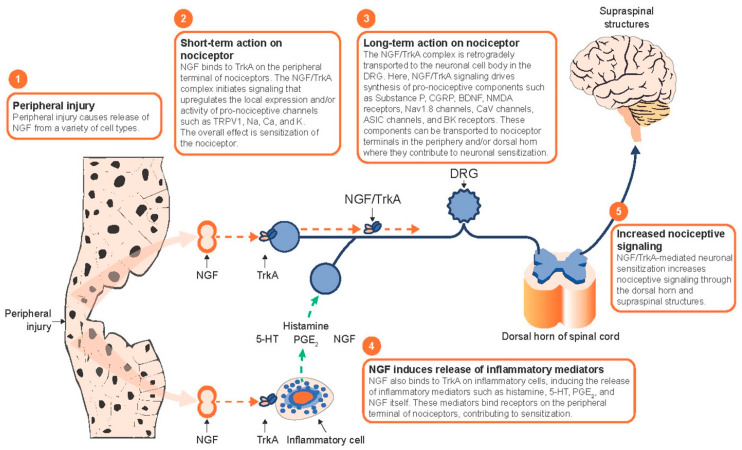
Role of NGF in stimulating peripheral nociceptor pain. Peripheral injury induces short- and long-term effects on nociceptors through the action of NGF, TrkA receptors, and inflammatory mediators. The result is increased nociceptive signaling. Modified from Barker et al., 2022 [148].

**Table 1 ijms-26-04663-t001:** Corneal nerves and their associated neurotransmitters [52].

Associated Corneal Nerve Type	Neurotransmitter
Sensory nerves	Substance P (SP)
	Calcitonin gene-related peptide (CGRP)
	Pituitary adenylate cyclase-activating peptide (PACAP)
	Galanin
	Excitatory amino acids (glutamate, aspartate)
Sympathetic nerves	Norepinephrine
	Neuropeptide Y
	Serotonin (5-HT)
Parasympathetic nerves	Vasoactive inhibitory peptide
	Met-enkephalin
	Neuropeptide Y
	Galanin
	Acetylcholine
Undetermined	Cholecystokinin
	Brain natriuretic peptide
	Vasopressin
	Neurotensin
	B-endorphin

**Table 2 ijms-26-04663-t002:** Current standard-of-care therapeutics for reducing corneal nociceptive pain and inflammation.

Type	Therapeutic	Mechanism	Effect
Broad-acting	NSAID	COX inhibitor reduces PGE2 release	Decreases nerve sensitization in PNS and CNS
	Acetaminophen	CNS	Decreases nerve sensitization
Specific	Bandage contact lenses	Direct physical barrier against mechanical irritants	Increases corneal healing and prevents chronic nociceptive pain
	Perfluorohexyl-octane (F6H8)	TRPM8 activator	Decreases nociceptive pain by improving dry eye symptoms; increases tear production and blinking rate
	Optive Plus artificial tears	TRPV1 antagonist containing L-carnitine	Decreases pain after 4 weeks

**Table 3 ijms-26-04663-t003:** Experimental therapeutics for modulation of corneal nociceptive pain and inflammation.

Stage	Therapeutic	Mechanism	Effect
Clinical trial—phase 2	Anti-TrkA monoclonal antibodies	TrkA inhibitor	Decreases nociceptive pain and inflammation
	Dual enkephalinase inhibitors	Inhibit enkephalin degradation, increasing opioid receptor binding	Reduces pain and inflammation
Pre-clinical	TrkA inhibitor	Monoclonal antibodies binding TrkA	Decreases nociceptive pain and sensitization in PNS and CNS
	SAF312, TRPV1 antagonist	TRPV1 selective, non-competitive antagonist, Ca^2+^ influx inhibitor	Inhibits Ca^2+^ influx into nociceptor cells, decreases inflammation
	Joint treatment with L-carnitine and capsazepine	TRPV1 antagonist, Ca^2+^ influx inhibitor	Reduces pain and discomfort
	Capsazepine, TRPV1, TRPV4, TRPM8 antagonist	Inhibits SP expression; Ca^2+^ influx inhibitor.	Inhibits Ca^2+^ influx into nociceptor cells, decreases corneal sensitization and inflammation
	TRPM8 ion channel antagonist	TRPM8 antagonist	Reduces inflammation
	DAMGO	Mu opioid receptor (MOR) ligand	Reduces responsiveness of nociceptors
	Aprepitant	Neurokinin 1 receptor (NK1R) antagonist	Decreases pain sensitivity and BDNF upregulation
	PIRT	Positive regulator of TRPV1 activity in nociceptive neurons	Influences pain perception, inflammation, and immune response; enhances nerve regeneration
	NGF	Promotes epithelial migration and proliferation	Improves would healing and nociceptor sensitivity
	Thy-1 YFP-positive myeloid-derived suppressor cells (MDSCs)	Producers of nerve growth factor (NGF)	Induces nociceptor growth
	Opioid growth factors	Analgesic	Supports wound healing and nociceptive sensitization

**Table 4 ijms-26-04663-t004:** Possible future targets for modulation of corneal nociceptive pain and inflammation.

Channel	Role in Nerve Recovery	Model	Source
TRPV4	Ablation of TRPV4 after nerve injury is related to the delay of nerve functional recovery	Mouse cell culture	[1]
TRPA1	Blocking TRPA1 results in decreased neuropathic pain in rat models	Rat	[3]
ASIC3	ASIC3 might improve tissue repair via a change in the M1: M2 macrophage ratio	Mouse	[4]
TRESK/TREK1	Overexpression of TRESK leads to faster mice paralysis recovery and lower TNF-α in blood	Mouse	[5]
PIEZO2	PIEZO2 is associated with mechanical allodynia after nerve injury	Mouse, human	[2,6]
PIRT	PIRT regulates TRPV1 with potential for nerve regeneration	Mouse	[7,8]

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
