# Peer review of "Corneal Sensory Receptors and Pharmacological Therapies to Modulate Ocular Pain"

_ijms, 2025, doi:10.3390/ijms26104663_

Round 1

Reviewer 1 Report

Comments and Suggestions for Authors

This is a interesting review paper explaining new receptors and substances concerning cornea nociceptor receptors and channel. I read on the paper many references overall in animal model or preclinical stage. So let s focus on the knowledge about human research.

P. Page 26 line 743-756 ;the NFG-based therapies including tanezumab and fasinumab antibodies against nerve growth factor, 1-¿The treatment for nociceptive pain in the cornea is administered via systemic or in topical formula?2- ¿do the authors could describe second effects of this drugs?

Page 29 table 4 the experimental therapeutics for modulation of corneal nociceptive pain and inflammation. There are four different family drugs that have different action mechanisms;3-¿which of this drugs have more powerful effect ? 4- ¿are any study that mix one or more substances?

Page 33 line 923-938 ; on page 34 table 5 we see that PIEZO2 is associated with mechanical allodynia after nerve injury in human model but the same authors said in the 938 line that current research on PIEZO2 therapeutics specific for corneal nociceptive is lacking,5-¿what is the meaning of that ?

The last comment goes to page 13 from line 337 to 342 , different studies speaking about corneal nerve damage after lasik in a high percentage of patients, however we see only a small amount of patients suffering from crónicas dry eye after this thecnique, 6¿this means that anatomical findings are not well correlated with clinical complaints ?.

Reviewer 2 Report

Comments and Suggestions for Authors

This is a very interesting review, but although I think that some sections are good and clear, some others are confusing and incomplete (overall sections 2 and 3). Also, classical and modern good publications or even book chapters (for example Adler’s physiology of the eye, chapter 16th) related with corneal nerves are missed, maybe this is the reason for the problems of some sections.

One important point is that the authors should use the functional classification of corneal sensory nerves (mechano-nociceptors, polymodal nociceptors, cold thermoreceptors) instead of the one depending on the presence of myelin (A-delta and C). Following the functional classification is easy to understand the channels (TRPs, Piezo…) they have on their membranes and how they are susceptible for being a pharmacological target. This can be found in some of the articles cited by the authors as Gerrero-Moreno et al. 2020 and Fakik et al. 2022, for example, but many other classical and modern works. I also think that is important not only talk about corneal nociceptors but also cold thermoreceptors. Both high and low threshold cold thermoreceptors as they are also involved in corneal-evoked pain (for example in dry eye disease) and therefore they are therapeutic targets (TRPM8 agonists and antagonists). Related with that, I am sure that nerves from figure 4 are cold thermoreceptors (TRPM8 positive). Also, I think that the word “pharmacological” should be added to the title, as many other therapies exist to treat pain. Following that my suggested title would be “Corneal sensory receptors and pharmacological therapies to modulate ocular pain

I am afraid that some important pain targets as coding channels are missed. I think the manuscript would be easier to understand if authors talk about transduction channels (TRPs, PIEZO, ASIC...) and coding channels (such as Kv, Nav channels). Potassium channels are mentioned in a paragraph within mechanoreception (section 2.2.3., lines 149-152) and this is not correct. And, what about Nav channels? what about Nav1.8 or Nav1.9 channels as key treatments?

Another important point is that I strongly encourage authors to review the text in detail, because there are certain explanations (overall in sections 2 and 3) that indicate serious conceptual errors that detract the work: for example, to confuse "sensation" with "stimuli" or "stimuli" with "evoked reflexes". The same happens with some explanations (as the one for the innervation of the cornea). I have detailed some of them later, but authors should review everything carefully and pay attention to these things.

I have also suggestions about the section’s organization, because from my point of view, they will clarify the manuscript:

- It would be better to explain first the anatomy of the innervation of the cornea and then the types of receptors, channels, etc.

- I think that for the purpose of the manuscript, section 3.2. "corneal nerve visualization" is unnecessary. In any case, it should also include also the existing experimental techniques, but I think this is not related with corneal pain treatment.

- Section 3.3. “Corneal nerve pathology” I think is incomplete as corneal lesion and neuropathies are missed. I think is that, instead this section, another general section talking about corneal nerves during inflammation and nerve lesion would be better. This new section could explain the general effects of corneal pathologies inducing pain and the mechanisms involved that are the target of the treatments.

- "nerve growth" is repeated in section 3.4. and 4.

- Some paragraphs are long-winded repetitive and unnecessary, such as lines 447-753. This section could easily start on line 454.

Other important general comments:

- When talking about the presence of transduction channels in the cornea, it is important to specify whether they have been found in the nerves or in another part of the corneal tissue (for example the epithelium). This is because many times it looks like the TRP channels are in the nerves, when they have only been described in the epithelium. If this is not clarified, it is confusing because sometimes the focus seems to be on nociceptors and other times not.

- I have detected a misinterpretation of the work cited as 58 (lines 439-440). In this work, a decrease in the mechanical threshold of the mechano-nociceptors was observed, but this indicates a sensitization of the nociceptors and therefore an increase in their response. This paragraph should be rewritten if this part of the work is kept.

- Please, pay attention to figure captions, they should be self-explained, and some of them are poorly explained or difficult to understand. Also, figure captions indicate "modified from..." but at least some of them have not been modified at all.

- The quality of many figures is poor, they look blurry.

- Some figures are completely unnecessary. Figure 13 is unnecessary, it has no meaning in the article and even less in the section on corneal nerve regeneration. Figure 14, Figure 15 and 16: What is the point of including figures on spinal nerve regeneration in a review of the cornea? Figure 19 is about remyelination, apparently not related directly with pain treatment and cornea (remember that corneal nerves loose myelin before entering the corneal tissue)

Specific comments:

- lines 54-55: “loss of nociceptor function can lead to chronic ocular pain”. This is not correct. It is not a loss of function but an alteration of nociceptor function that leads to chronic ocular pain.

- lines 57-58: If the authors are referring to allodynia, this is not exactly the case.

- lines 182-185: this paragraph is confusing, it must be rewritten because the innervation of the cornea is not well described. It says “the cornea contains ophthalmic branches…”, it should say “the cornea is innervated by the ophthalmic branches…”, and things like that.

- line 189-190: “These corneal nerve fibers can respond to many sensations and other stimuli (blinking, tear production)”. Nociceptors do not respond to “sensations” they respond to stimuli; blinking and tear production are not stimuli, are reflexes evoked by the corneal nerve activity. Please review carefully all the text.

- line 736-737: should it says “the activation of TRMP8 also increases the expression of Tac1”?

- table 1 is not cited in the text. This table must be full revised if the authors want to keep it. For example: the table is a compendium about some transduction channels and their principal activators and others. What does the title “nociceptors studied in relation to nerve injury” means? The primary function of TRPV1 is to detect “pain”? Pain is not a stimulus, is a sensation…What about the coding channels (Kv, Nav…) in this table?

- table 2. The title should be the other way around: corneal nerves and their associated neurotransmitter…

- Section 3.3.1.: dry eye syndrome? or dry eye disease (DED)? DED is more accepted as is considered more precise and complete.

- Saying that F6H8 is an analgesic is not correct since its mechanism of action, as far as is currently known, is not to block the activity of a nociceptor, but rather to alleviate dry eye pathology by other mechanisms.

- Figure 18 caption, the authors of the original article are missed.

- Table 5 is uncompleted for PIRT

- What about the future treatments with TRPM8 agonists and antagonists? Add them in section 6.1.1.

Reviewer 3 Report

Comments and Suggestions for Authors
  1. The quality of English in this section is generally clear and well-structured, but there are areas that could be improved for better readability and academic precision. Some sentences are overly complex, making it difficult to follow the main ideas. Simplifying sentence structures and ensuring clarity in explanations would enhance readability. Additionally, there are instances of awkward phrasing, redundancy, and occasional grammatical inconsistencies that could be refined for smoother flow.
  2. Furthermore, the integration of figures and references sometimes lacks seamless transitions, making it harder for readers to connect the visual data with the textual discussion. A more cohesive link between figures and their explanatory content would improve the overall clarity. Minor issues with subject-verb agreement and article usage (such as missing "the" before nouns) are present and should be addressed.
  3. Overall, while the language is comprehensible, refining sentence structure, reducing redundancy, and improving clarity in figure descriptions would enhance the readability and impact of the section. Proofreading by a native or proficient English speaker or using professional language editing services would be beneficial.
  4. One major weakness of the section is the lack of direct experimental evidence linking the discussed TRP channels to specific nociceptive mechanisms in the cornea. While TRPV1, TRPM8, and TRPA1 are well-documented in pain perception, the roles of TRPC4 and TRPV4 in corneal nociception are primarily speculative. The text does not provide sufficient in vivo studies or functional assays that confirm the direct involvement of these channels in corneal pain signaling, making some of the claims less substantiated.
    Another limitation is the insufficient discussion of cross-talk between different nociceptive channels. The text briefly mentions the interaction between TRPV1 and TRPM8, but other potential interactions, such as those between TRPA1 and TRPV1 or TRPM8 and ASICs, are not explored. Given that TRPV1 and ASICs both respond to acidic conditions, their possible functional interplay in corneal pain should have been examined. A deeper discussion on how these channels influence each other’s activation, particularly in sensitization processes, would strengthen the paper.
  5. The section also lacks a detailed explanation of the molecular mechanisms underlying TRP channel sensitization in corneal pain. While it states that persistent activation of these channels leads to nociceptive pain, it does not delve into the phosphorylation events, second messenger pathways, or regulatory protein interactions that contribute to channel sensitization. This omission leaves a gap in understanding how these channels transition from normal function to a hyperactive pain state in chronic ocular conditions.
  6. Additionally, the discussion on TRP channel localization in the cornea is somewhat superficial. While the figures highlight TRPM8 clustering in nociceptors, there is no comparative analysis of whether TRPV1, TRPV4, or TRPA1 exhibit similar or different spatial distributions. Understanding the localization of these channels is crucial for interpreting their functional roles in corneal pain perception, yet the section does not address these nuances.
  7. A methodological issue in this section is its reliance on data derived from non-human species, such as guinea pigs. While findings from animal models can be informative, the direct applicability of these results to human corneal nociception is unclear. The section does not discuss potential species-specific differences in TRP channel expression, which could impact the validity of these findings in a clinical setting. This omission weakens the translational relevance of the discussion.
    Another limitation is the potential overinterpretation of immunohistochemistry data. The section claims that TRPM8 fibers form unique clusters in the corneal epithelium, but it does not provide functional evidence that this clustering has a specific effect on nociceptive signaling. Without electrophysiological recordings or behavioral pain response studies, it is unclear whether TRPM8 clustering leads to distinct sensory properties, such as increased cold sensitivity or enhanced pain signaling.
  8. One of the major weaknesses of this section is the lack of discussion on how the described TRP channels could be targeted for therapeutic interventions in human ocular pain conditions. While the text provides an extensive catalog of TRP channel types and their general functions, it does not translate this knowledge into clinical applications. There is no mention of ongoing drug development efforts aimed at modulating these channels for pain relief, which limits the practical implications of the findings.
  9. Another gap in the discussion is the absence of consideration for variability in patient populations. The section does not address how age-related changes might affect TRP channel expression and corneal pain perception, which is particularly important since corneal pain syndromes often affect older adults. Additionally, potential sex differences in TRP channel function, which have been reported in pain research, are not explored. A more comprehensive analysis that includes these factors would improve the applicability of the findings.
  10. The paper relies heavily on figures but does not provide sufficient explanation or critical analysis of their relevance to corneal nociception. Many figures, such as those adapted from Clapham (2003) and Yang et al. (2018), primarily depict general TRP channel properties rather than their specific roles in corneal pain. Without deeper analysis of these figures, their inclusion does not significantly enhance the reader’s understanding.
    Additionally, the paper lacks a clear research question or hypothesis, making it feel more like a general catalog of TRP channels rather than a focused scientific discussion. Instead of simply listing different TRP channel families and their properties, the text should emphasize which channels are most relevant for corneal nociception and which gaps in knowledge need to be addressed. A stronger focus on key research questions would improve the clarity and impact of the section.
Comments on the Quality of English Language

The quality of English in this section is generally clear and well-structured, but there are areas that could be improved for better readability and academic precision. Some sentences are overly complex, making it difficult to follow the main ideas. Simplifying sentence structures and ensuring clarity in explanations would enhance readability. Additionally, there are instances of awkward phrasing, redundancy, and occasional grammatical inconsistencies that could be refined for smoother flow.

Furthermore, the integration of figures and references sometimes lacks seamless transitions, making it harder for readers to connect the visual data with the textual discussion. A more cohesive link between figures and their explanatory content would improve the overall clarity. Minor issues with subject-verb agreement and article usage (such as missing "the" before nouns) are present and should be addressed.

Round 2

Reviewer 2 Report

Comments and Suggestions for Authors

The manuscript has been greatly improved thanks to the reviewers' recommendations, and the authors’ effort. I think the organization is now more understandable and the content more complete. I still have detected some errors that, if corrected, will be improve the manuscript.

- The sentence from lines 66 to 68 is not correct, it should be something like that “the central branch of the trigeminal neurons travels to the trigeminal brainstem sensory complex in the pons, while the peripheral branch that innervates the ipsilateral cornea travels into the orbit through the ophthalmic branch of the trigeminal ganglion to provide the sensory supply to the ipsilateral cornea”. This is clearly shown in figure 1!!!

- Part of the sentence from lines 550 to 552 is not correct. Corneal inflammation (as in UV photokeratitis or allergic keratoconjunctivitis) induces sensitization (increase in the response) of mechano- and polymodal nociceptors and desensitization (decrease in the response) of cold thermoreceptors due to the release of inflammatory mediators. For example, it is known that the inflammatory mediators inhibit TRPM8 channels through a specific protein (Zhang et al 2012). So, I would change the sentence like this: “Despite increase sensitivity to mechanical and chemical noxious stimuli, decreased sensitivity to other stimuli occurs, implying…”

- Paragraph from lines 784 to797 is repeated (is exactly the same as from lines 728 to 741), It has to be deleted (delete lines 784 to 797).

- I have detected that the citation number 24 in point 7.1.5 is not about aprepitant. Please, review all the bibliography, because some citations could be missed of moved…

- review the spelling; a symbol has been inserted instead of "delta" for the A-delta fibers.

- review the spelling of citations number 4 and 54

- line 361: I think you mean “brainstem” instead of “brain”

Other comments:

- I think that the sentence from lines 61 to 63 has no sense in this part of the manuscript. I would delete it.

Reviewer 3 Report

Comments and Suggestions for Authors

While the paper comprehensively catalogs corneal nociceptive pathways, ion channels, and therapeutic interventions, it lacks a cohesive synthesis that interrelates these components into a conceptual framework. The manuscript reads more as an encyclopedia of facts than a critical review that guides future research directions. A more structured model of how specific molecular pathways interact in the context of disease and recovery (e.g., dry eye, post-surgical pain) would enhance both scientific utility and translational relevance.

Much of the experimental evidence referenced is derived from rodent models, particularly in discussions of TRP channel expression and function. While these models are essential, the lack of human corneal-specific data (e.g., from clinical samples, imaging, or human trials) undermines the clinical applicability of several conclusions. The authors acknowledge this in part but do not systematically differentiate between species-specific mechanisms and conserved features.

The review rarely discusses studies with conflicting findings or inconclusive results. For instance, TRPA1’s role in cold sensing is described with ambiguity, but the narrative does not explore conflicting physiological interpretations or implications for drug targeting. A more nuanced treatment of controversies would reflect a mature scientific perspective and guide future hypothesis testing.

The section on emerging therapeutics, including TRP antagonists and NGF inhibitors, is extensive but lacks a critical evaluation of their current clinical limitations, side effects, or barriers to translation. The discussion would benefit from a structured table comparing efficacy, delivery methods, trial phases, and known toxicities. Additionally, the dual role of TRP channels in nociception and regeneration is not adequately balanced in discussions of potential inhibitors, which may have adverse effects on healing.

Although the manuscript introduces sex- and age-related variations in TRP channel function, this section remains underdeveloped. The biological significance and potential for sex-specific treatment strategies are not thoroughly discussed, nor is there a proposal for stratified therapeutic design. This is a missed opportunity to advance personalized medicine perspectives.
The review focuses heavily on acute pain mechanisms and short-term interventions but gives minimal attention to chronic ocular pain syndromes and their neuroplastic changes. Corneal neuropathic pain, which can persist post-refractive surgery or in chronic dry eye, is underexplored in terms of both pathophysiology and therapeutic innovation。

There is little mention of how advancements in in vivo confocal microscopy, tear biomarkers, or electrophysiological tools could complement the understanding or monitoring of corneal pain. Integration of diagnostic methodologies could add translational depth to the molecular and therapeutic discussion.

The paper includes exhaustive lists of neurotransmitters, ion channels, and cytokines without prioritizing their clinical relevance or degree of supporting evidence. This can overwhelm readers and obscure the most promising therapeutic targets. A tiered presentation based on translational readiness would improve readability.

While some combination therapies (e.g., capsazepine + L-carnitine) are mentioned, the review does not broadly explore the promise and challenges of multimodal pain management strategies or delivery systems such as sustained-release implants, hydrogels, or gene therapy.
The manuscript lacks any discussion on the regulatory landscape (e.g., challenges in getting ocular analgesics approved), or the economic feasibility and accessibility of advanced therapies. These are crucial aspects for translating experimental findings into widespread clinical practice.

While the paper provides a thorough review of the molecular mechanisms and therapeutic approaches for corneal nociception, future research would benefit from integrating artificial intelligence (AI) to enhance data interpretation and therapeutic development. AI-driven bioinformatics and machine learning models can be employed to analyze large-scale omics data, correlate ion channel expression patterns with clinical outcomes, and identify novel drug targets or biomarkers for ocular pain. Additionally, AI-powered image analysis of in vivo confocal microscopy or corneal nerve imaging could allow for more precise and automated evaluation of nerve regeneration, morphology, and therapeutic responses. Moving forward, interdisciplinary collaboration that combines computational modeling with experimental and clinical ophthalmology will be essential to accelerate personalized and predictive pain management strategies in corneal diseases.

Comments on the Quality of English Language

The manuscript is generally well-written and demonstrates a strong command of scientific English. Terminology is appropriately used, and the sentence structures are clear and academically formal. However, the text would benefit from minor language polishing to improve readability and flow, particularly in areas with dense technical descriptions or long compound sentences. Occasional redundancies and awkward phrasing could be refined for greater clarity and conciseness. Overall, the language quality is sufficient for publication but could be enhanced with light editorial revision.
